# IDEAL: Interpretable-by-Design ALgorithms for learning from foundation feature spaces

## Abstract

Many of the existing transfer learning methods rely on parametric tuning and lack interpretation of decision making. However, the advance of foundation models (FM) makes it possible to avoid such parametric tuning, taking advantage of pretrained feature spaces. In this study, we define a framework called IDEAL (Interpretable-by-design DEep learning ALgorithms) which tackles the problem of interpretable transfer learning by recasting the standard supervised classification problem into a function of similarity to a set of prototypes derived from the training data. This framework generalises previously-known prototypical approaches such as ProtoPNet, xDNN and DNC. Using the IDEAL approach we can decompose the overall problem into two inherently connected stages: A) feature extraction (FE), which maps the raw features of the real-world data into a latent space, and B) identification of representative prototypes and decision making based on similarity and association between the query and the prototypes. This addresses the issue of interpretability (stage B) while retaining the benefits from the tremendous achievements offered by deep learning (DL) models (e.g., visual transformers, ViT) which are often pre-trained on huge datasets such as IG-3.6B + ImageNet-1K or LVD-142M (stage A).

On a range of datasets (CIFAR-10, CIFAR-100, CalTech101, STL-10, Oxford-IIIT Pet, EuroSAT), we demonstrate, through an extensive set of experiments, how the choice of the latent space, prototype selection, and finetuning of the latent space affect accuracy and generalisation of the models on transfer learning scenarios for different backbones. Building upon this knowledge, we demonstrate that the proposed framework helps achieve an advantage over state-of-the-art baselines in class-incremental learning. Finally, we analyse the interpretations provided by the proposed IDEAL framework, as well as the impact of confounding in transfer learning, demonstrating that the proposed approach **without** finetuning improves the performance on confounded data over finetuned counterparts.

The key findings can be summarised as follows: (1) the setting allows interpretability through prototypes, while also mitigating the issue of confounding bias, (2) lack of finetuning helps circumvent the issue of catastrophic forgetting, allowing efficient class-incremental transfer learning, and (3) ViT architectures narrow the gap between finetuned and non-finetuned models allowing for transfer learning in a fraction of time **without** finetuning of the feature space on a target dataset with iterative supervised methods.

## 1    Background

Deep-learning (DL) models can be formulated as deeply embedded functions of functions (Angelov & Gu (2019), Rosenblatt et al. (1962)), optimised through backpropagation (Rumelhart et al. (1986)):

$$\hat{y}(\mathbf{x}) = f_n(\dots (f_1(\mathbf{x}; \boldsymbol{\theta}_1) \dots); \boldsymbol{\theta}_n), \tag{1}$$

where $f_n(\dots (f_1(\mathbf{x}; \boldsymbol{\theta}_1) \dots); \boldsymbol{\theta}_n)$ is a layered function of the input $\mathbf{x}$, which has a generic enough, fixed parameterisation $\boldsymbol{\theta}$. to predict desirable outputs $\hat{y}$.

However, this problem statement has the following limitations:

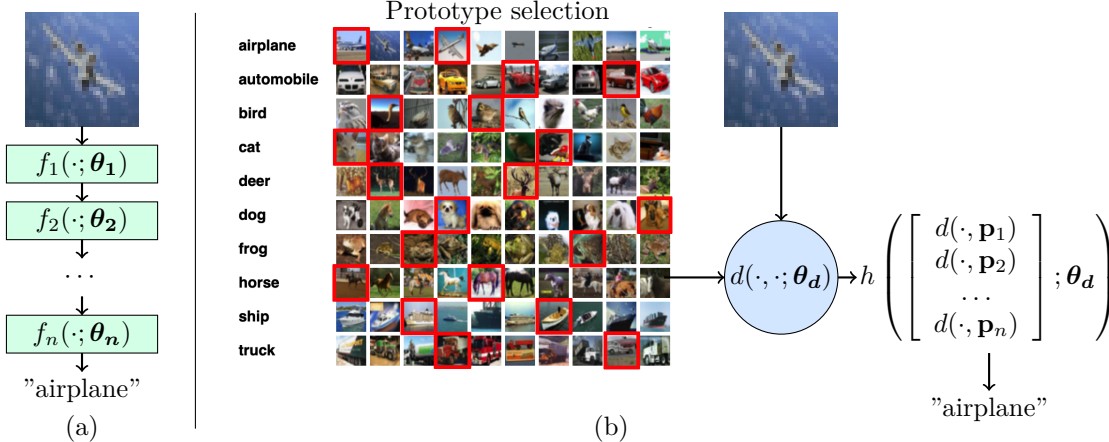

Figure 1: Difference between (a) a standard deep-learning model, and (b) the proposed prototype-based approach, IDEAL. Dataset credit: CIFAR-10 (Krizhevsky & Hinton (2009))

(1) transfer learning typically requires finetuning (Kornblith et al. (2019)) using error back-propagation (EBP) on the target problem and data of interest

(2) such formulation does not depend upon training data, so the contribution of these samples towards the output $\hat{y}$ is unclear, which hinders interpretability. For the interpretable architectures, such as ProtoPNet (Chen et al. (2019)), finetuning leads to confounding interpretations (Bontempelli et al. (2022))

(3) finally, for lifelong learning problems, such finetuning creates obstacles such as catastrophic forgetting (Parisi et al. (2019))

Emergence of foundation models, aimed at better generalisation and facilitating transfer learning, allows for mitigating point (1). Studies such as DINOv2 (Oquab et al. (2023)) demonstrate competitive results for ViT-based (Dosovitskiy et al. (2020)) architectures on transfer learning tasks on a range of datasets with linear finetuning. However, these works do not address neither interpretability nor lifelong learning. In this work, to jointly address all three above-mentioned limitations, we propose a generic framework for prototypical transfer learning called IDEAL (Interpretable-by-design DEep learning ALgorithms). Through this framework, we extensively study the benefits and trade-offs of prototypical transfer learning without finetuning across different architectures and tasks.

Our solution for transfer learning, which generalises of xDNN (Angelov & Soares (2020)) and ProtoPNet (Chen et al. (2019)), can be summarised in the following form:

$$\hat{y} = g(\mathbf{x}; \boldsymbol{\theta}, \mathbb{P}), \tag{2}$$

where $\mathbb{P}$ is a set of prototypes. We consider a more restricted version of function $g(\cdot)$:

$$\hat{y} = g(\mathbf{x}; \boldsymbol{\theta}_{\{d,h\}}, \mathbb{X}) = h(d(\mathbf{x}, \mathbf{p}; \boldsymbol{\theta}_d)|_{\mathbf{p} \in \mathbb{P}}; \boldsymbol{\theta}_h), \tag{3}$$

where $d$ is some form of (dis)similarity function (which can include DL feature extractors), $\boldsymbol{\theta}_d$ and $\boldsymbol{\theta}_h$ are parameterisations of functions $d$ and $h$, respectively.

The methods from this research draw from cognitive science and the way humans learn, namely using examples of previous observations and experiences (Zeithamova et al. (2008)). Prototype-based models have long been used in different learning systems: $k$ nearest neighbours (Radovanovic et al. (2010)); decision trees (Nauta et al. (2021)); rule-based systems (Angelov & Zhou (2008)); case-based reasoning (Kim et al. (2014)); sparse kernel machines (Tipping (1999)). The advantages of prototype-based models has been advocated, for example, in Bien & Tibshirani (2011). The first prototypical architecture, learning both distances and prototypes, was proposed in Snell et al. (2017) and more recently developed in Chen et al. (2019); Angelov & Soares (2020) and Wang et al. (2023).

In this paper, we demonstrate the efficiency of the proposed framework: it is compact, easy to interpret by humans, fast to train and adapt in a lifelong learning setting and benefits from a latent data space learnt from a generic dataset transferred to a different, more specific domain.

Specifically, we make the following contributions:

- we define the framework called IDEAL, which transforms a given non-interpretable latent space into an interpretable one based on prototypes, derived from the training set without finetuning, and quantify the performance gap between such model, its finetuned counterpart and standard DL architectures.

- we demonstrate the benefits of the proposed framework on transfer and lifelong learning scenarios. Namely, in a fraction of training time and **without finetuning** of latent features the proposed models achieve performance, competitive with standard DL techniques.

- we demonstrate the model's interpretability on classification and lifelong learning tasks, and show that **without** finetuning, the resulting models achieves **better** performance on confounded CUB data comparing to finetuned counterparts (Wah et al. (2011); Bontempelli et al. (2022))

We apply this generic IDEAL framework to a set of standard DL architectures such as ViT (Dosovitskiy et al. (2020); Singh et al. (2022)), VGG (Simonyan & Zisserman (2014)), ResNet (He et al. (2016)) and xDNN (Angelov & Soares (2020)) and evaluate the methodology on a range of well-known datasets such as CIFAR-10, CIFAR-100, CalTech101, EuroSAT, Oxford-IIIT Pet, and STL-10.

## 2 Related work

**Explainability and interpretability**  The ever more complicated DL models (Krizhevsky et al. (2012); Dosovitskiy et al. (2020)) do not keep pace with the demands for human understandable interpretability (Rudin (2019)). Interpretability of deep neural networks is especially important in a number of applications: automotive (Kim & Canny (2017)), medical (Ahmad et al. (2018)), Earth observation (Zhang et al. (2022)) alongside others. Demand in such models is necessitated by the pursuit of safety (Wei et al. (2022)), as well as ethical concerns (Peters (2022)). Some of the pioneering approaches to explaining deep neural networks involve *post hoc* methods; these include saliency models such as saliency map visualisation method (Simonyan et al. (2014)) as well as Grad-CAM (Selvaraju et al. (2017)). However, saliency-based explanations may be misleading and not represent the causal relationship between the inputs and outputs (Atrey et al. (2019)), representing instead the biases of the model (Adebayo et al. (2018)).

An arguably better approach is to construct interpretable-by-design (*ante hoc*) models (Rudin (2019)). These models could use different principles such as (1) interpretable-by-design architectures (Böhle et al. (2022)), which are designed to provide interpretations at every step of the architecture, as well as (2) prototype-based models, which perform decision making as a function of (dis)similarity to existing prototypes (Angelov & Soares (2020)). One of the limitations of the prototype based methods is that they are often still based on non-interpretable similarity metrics. This can be considered an orthogonal open problem which can be addressed by providing interpretable-by-design DL architectures (Böhle et al. (2022)).

**Symbolic and sparse learning machines**  The idea of prototype-based machine learning is closely related to the symbolic methods (Newell et al. (1959)), and draws upon the case based reasoning (Kim et al. (2014)) and sparse learning machines (Poggio & Girosi (1998)), which are designed to learn a linear (with respect to parameters) model, which is (in general, nonlinearly) dependent on a subset of training data samples. At the centre of many such methods is the kernel trick (Schölkopf et al. (2001)), which involves mapping of training and inference data into a space with different inner product within a reproducing Hilbert space (Aronszajn (1950)). Such models include support vector machines (SVMs) for classification (Boser et al. (1992)) and support vector regression (SVR) models (Smola & Schölkopf (2004)) for regression, as well as relevance vector machines (RVMs), which have demonstrated improvements in sparsity (Tipping (2001)).

**Prototype-based models** (Snell et al. (2017)) proposed to use a single prototype per class in a few-shot supervised learning scenario. Another study by Li et al. (2018) suggested prototype-based learning for interpretable case-based reasoning. Building upon it, (Chen et al. (2019)) developed ProtoPNet model which classifies an image through dissecting it into a number of patches, which are then compared to prototypes for decision making using end-to-end supervised training. A more recent model xDNN (Angelov & Soares (2020)) selects one or multiple prototypes per class through a non-iterative online procedure which uses data density. In contrast to the proposed setting, it uses finetuning on a downstream dataset and only uses weak backbone models such as VGG-16.

Versions of xDNN also define prototypes at the level of segments (Soares et al. (2021)) and image pixels (Zhang et al. (2022)). The concept of xDNN was used in the end-to-end prototype-based learning method DNC (Wang et al. (2023)). In contrast to xDNN and DNC, we consider the **lifelong learning** scenario and investigate the properties of models, trained on generic and **not finetuned** datasets.

The closest works to this study are prototype-based models ProtoPNet (Rudin (2019)), DNC (Wang et al. (2023)) and xDNN (Angelov & Soares (2020)). In fact, the proposed framework generalises these methods as shown in Section 3.3. These works, however, are focused on end-to-end training and are not motivated by the challenge of transfer learning.

**Large deep-learning classifiers** In contrast to DNC (Wang et al. (2023)) and ProtoPNet (Chen et al. (2019)), the proposed framework goes beyond the end-to-end learning concept. Instead, it takes advantage of the feature space of large classifiers such as ResNet (He et al. (2016)), VGG (Simonyan & Zisserman (2014)), SWAG-ViT (Singh et al. (2022)), and shows that with carefully selected prototypes one can achieve, on a number of datasets, a performance comparable to end-to-end trained models, in offline and online (lifelong) learning scenarios with or even **without finetuning and end-to-end learning**, thus very fast and computationally efficient, yet interpretable.

**Continual learning** Continual learning models solve a number of related problems (van de Ven et al. (2022)). *Task-incremental learning* addresses the problem of incrementally learning known tasks, with the intended task explicitly input into the algorithm (Ruvolo & Eaton (2013); Li & Hoiem (2017); Kirkpatrick et al. (2017)). *Domain-incremental learning* (Wang et al. (2022a); Lamers et al. (2023)) addresses the problem of learning when the domain is changing and the algorithm is not informed about these changes. This includes such issues as *concept drift* when the input data distribution is non-stationary (Widmer & Kubat (1996)). *Class-incremental learning* (Yan et al. (2021); Wang et al. (2022b)) is a problem of ever expanding number of classes of data. In this paper, we only focus on this last problem. However, one can see how the prototype-based approaches could help solve the other two problems by circumventing catastrophic forgetting (French (1999)) through incremental update of the prototypes (Baruah & Angelov (2012)).

**Clustering** Critically important for enabling continual learning is to break the iterative nature of the end-to-end learning and within the proposed concept which offers to employ clustering to determine prototypes. Therefore, we are using both online (ELM (Baruah & Angelov (2012)), which is an online version of mean-shift (Comaniciu & Meer (2002))), and offline $k$-means (MacQueen et al. (1967)) methods. Although there are a number of online clustering methods, e.g. the stochastic Chinese restaurant process Bayesian non-parametric approach (Aldous et al. (1983)), they usually require significant amount of time to run and therefore we did not consider those.

## 3 Methodology

### 3.1 Problem statement

Two different definitions of the problem statement are considered: offline and online (lifelong) learning.

**Offline learning**   Consider the following optimisation problem:

$$\arg\min_{\substack{\mathbb{P}=\mathbb{P}(\mathbb{X}),\\ \boldsymbol{\theta}_{\{d,h\}}}} \sum_{(\mathbf{x},y)\in(\mathbb{X},\mathbb{Y})} l(h(d(\mathbf{x},\mathbf{p};\boldsymbol{\theta}_d)|_{\mathbf{p}\in\mathbb{P}};\boldsymbol{\theta}_h),y), \tag{4}$$

where $(\mathbb{X},\mathbb{Y})$ are a tuple of inputs and labels, respectively, and $\mathbb{P}$ is a set of prototypes derived from data $\mathbb{X}$ (e.g., by selecting a set of representative examples or by clustering).

Brute force optimisation for the problem of selecting a set of representative examples is equivalent to finding a solution of the best-subset selection problem, which is an NP-hard problem (Natarajan (1995)). While there are methods for solving such subset selection problems in limited cases such as sparse linear regression (Bertsimas et al. (2016)), it still remains computationally inefficient in a general case (polynomial complexity is claimed in Zhu et al. (2020)) and/or solving it only in a limited (i.e. linear) setting.

The common approach to dealing with such selection problem is to replace the original optimisation problem (equation (4)) with a surrogate one, where the prototypes $\mathbb{P}$ are provided by a data distribution (Angelov & Soares (2020)) or a geometric, e.g. clustering (Wang et al. (2023)) technique. Then, once the prototypes are selected, the optimisation problem becomes:

$$\arg\min_{\boldsymbol{\theta}_{\{d,h\}}} \sum_{(\mathbf{x},y)\in(\mathbb{X},\mathbb{Y})} l(h(d(\mathbf{x},\mathbf{p};\boldsymbol{\theta}_d)|_{p\in\mathbb{P}};\boldsymbol{\theta}_h),y). \tag{5}$$

**Online (lifelong) learning**   Instead of solving a single objective for a fixed dataset, the problem is transformed into a series of optimisation problems for progressively growing set $\mathbb{X}$:

$$\{\arg\min_{\boldsymbol{\theta}_{\{d,h\}}} \sum_{(\mathbf{x},y)\in(\mathbb{X}_n,\mathbb{Y}_n)} l(h(d(\mathbf{x},\mathbf{p};\theta_d)|_{\mathbf{p}\in\mathbb{P}_n};\boldsymbol{\theta}_h),y)\}_{n=1}^N, \mathbb{X}_n=\mathbb{X}_{n-1}+\{\mathbf{x}_n\}, \mathbb{X}_1=\{\mathbf{x}_1\}. \tag{6}$$

Once the prototypes are found, the problem would only require light-weight optimisation steps as described in Algorithms 1 and 2.

---

**Algorithm 1:** Training and testing (offline)

---

**Data:** Training data $\mathbb{X}=\{\mathbf{x}_1\ldots\mathbf{x}_N\}$;
**Result:** Prototype-based classifier $c(\mathbf{x};\mathbb{P},\boldsymbol{\theta})$
$\mathbb{P}\leftarrow \texttt{FindPrototypes}(\{\mathbf{x}_1\ldots\mathbf{x}_N\})$;    // Prototype selection function $\texttt{FindPrototypes}:\mathbb{X}\to\mathbb{P}$
$\theta\leftarrow \texttt{SelectParameters}(\mathbb{X},\mathbb{Y},\boldsymbol{\theta})$;        // $\texttt{SelectParameters}$ is a solution of Eq. 5
$\hat{\mathbb{Y}}_T\leftarrow \{h(d(\mathbf{x},\mathbf{p};\boldsymbol{\theta}_d)|_{\mathbf{p}\in\mathbb{P}};\boldsymbol{\theta}_h)\}_{\mathbf{x}\in\mathbb{X}_T}$;

---

**Algorithm 2:** Training and testing (online)

---

**Data:** Training data $\mathbb{X}=\{\mathbf{x}_1\ldots\mathbf{x}_N\}$;
**Result:** Prototype-based classifier $h(d(\mathbf{x},\mathbf{p};\boldsymbol{\theta}_1)|_{p\in\mathbb{P}};\boldsymbol{\theta}_2)$
$\mathbb{P}\leftarrow\{\}$;
**for** $\{\mathbf{x},y\}\in\mathbb{X}$ **do**
  $\hat{y}=h(d(\mathbf{x},\mathbf{p};\boldsymbol{\theta}_d)|_{\mathbf{p}\in\mathbb{P}};\boldsymbol{\theta}_h)$;
  $\mathbb{P}\leftarrow \texttt{UpdatePrototypes}(\mathbb{P},\mathbf{x})$;   // Prototype update function $\texttt{UpdatePrototypes}:\mathbb{P}\times\mathbb{X}\to\mathbb{P}$
  $\theta\leftarrow \texttt{UpdateParameters}(\mathbb{X},\mathbb{Y},\boldsymbol{\theta})$;           // $\texttt{UpdateParameters}$ is a solution of Eq. 6
**end**

---

## 3.2   Choice of functions $d$ and $h$

While we define the framework in generic terms, we limit our analysis to a special case of Euclidean distance and winner-takes-all function. This helps focus on quantifying the trade-offs of accuracy, interpretability and generalisation between the model without finetuning, on one hand, and state-of-the-art, fully finetuned, models. Although it may be possible to further improve the performance by finding better architectural

choices, we decided to focus on the simple parameterisation of the framework with the Euclidean distance and a winner-takes-all decision making.

Throughout the experiments, we use the negative Euclidean distance between the feature vectors:

$$d(\mathbf{x}, \mathbf{p}; \theta_d) = -\ell^2(\phi(\mathbf{x}; \theta_d), \phi(\mathbf{p}; \theta_d)), \tag{7}$$

where $\phi$ is the feature extractor output. For the scenario without finetuning, $\theta_d$ is frozen: $\phi(\cdot) = \phi(\cdot; \theta_d), \theta_d = $ const. The similarities bounded between $(0, 1]$ could be obtained by, for example, taking the exponential of the similarity function or normalising it. Except from the experiment in Figure 4, where $h$ is implemented as $k$-NN, the function $h$ is a winner-takes-all operator:

$$h(\cdot) = \text{CLASS}(\arg \min_{p \in \mathbb{P}} d(\cdot, \mathbf{p}; \theta_d)) \tag{8}$$

Note that the lack of finetuning makes the loss function trivial as the model does not have any free parameters $\theta_{\{d,h\}}$.

### 3.3 Difference from the other prototype-based frameworks

Existing prototype-based models, such as ProtoPNet (Chen et al. (2019)), DNC (Wang et al. (2023)) and xDNN (Angelov & Soares (2020)), focus on end-to-end training for the purpose of interpretability by design and not on transfer learning from the existing pretrained models. All of them can also be considered as specific cases of the presented framework. Neither of these models are aiming to address transfer learning, in contrast to this paper's attention on the trade-offs between finetuned and non-finetuned models.

**xDNN** (Angelov & Soares (2020)) is a special case of our IDEAL formulation with

$$d(\mathbf{x}, \mathbf{p}; \theta_d) = -\mathfrak{C}(\phi(\mathbf{x}; \theta_d), \phi(\mathbf{p}; \theta_d)), \tag{9}$$

where $\mathfrak{C}$ is a Cauchy similarity. It optimises the coefficients $\theta_d$ as a part of its finetuning procedure prior to the model training, and its decision making is defined according to the winner-takes-all procedure as per Equation 8.

**DNC** (Wang et al. (2023)) selects prototypes at every optimisation step using an online version of Sinkhorn-Knopp clustering algorithm (Cuturi (2013)) and defines $l$ in equation 6 a softmax cross-entropy loss.

**ProtoPNet** (Chen et al. (2019)), in contrast to the former two methods, operates over patches and not the full images:

$$d(\mathbf{x}, \mathbf{p}; \theta_d) = \max_{\hat{\mathbf{x}} \in \text{patches}(\mathbf{x})} (\log(\ell^2(\hat{\mathbf{x}}, \mathbf{p}) + 1) - \log(\ell^2(\hat{\mathbf{x}}, \mathbf{p}) + \epsilon), \tag{10}$$

where $\epsilon$ is a parameter. ProtoPNet also defines a decision making function $h$ as follows and optimises jointly the prototypes $\mathbb{P}$ and the parameters $\theta_{d,h}$ using cross-entropy loss:

$$h(\cdot) = \text{FC}\left(\begin{bmatrix} d(\cdot, p_1) \\ d(\cdot, p_2) \\ \dots \\ d(\cdot, p_{|\mathbb{P}|}) \end{bmatrix}_{p_i \in \mathbb{P}, i \in [1, \dots, |\mathbb{P}|]}\right) \tag{11}$$

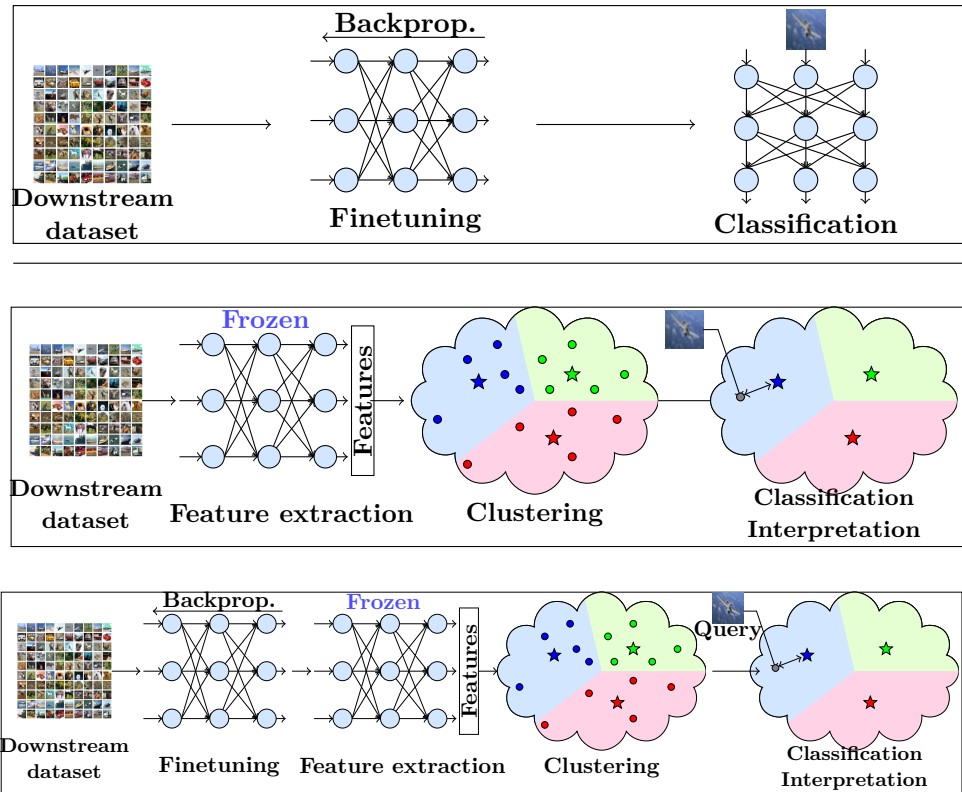

Figure 2: Experimental setup. Top: standard DL model; middle: proposed framework with **no** finetuning; bottom: proposed framework **with** finetuning

### 3.4 Prototype selection through clustering

Selection of prototypes through many standard methods of clustering, such as $k$-means (Steinhaus et al. (1956)), is used by methods such as (Zhang et al. (2022)), DNC (Wang et al. (2023)). However, these methods have one serious limitation: they utilise the averaging of cluster values, so the prototypes $\mathbb{P}$ do not, in general, belong to the original training dataset $\mathbb{X}$. It is still possible, however, to attribute the prediction to the set of the cluster members.

The possible options for such prototype selection are summarised below. Standard *black-box* classifiers do not offer interpretability through prototypes. Prototypes, selected through $k$-means, are non-interpretable on their own account as discussed above; however, it is possible to attribute such similarity to the members of the clusters. Finally, one can select real prototypes as cluster centroids. This way it is possible to attribute the decision to a number of real image prototypes ranked by their similarity to the query image. Such choice between averaged and real centroids can create, as we show in the experimental section, a trade-off between interpretability and performance (see Section 4.2).

## 4 Experiments

Throughout the experimental scenarios, we contrast three settings (see Figure 2):

A) Standard DL pipeline involving training on generic datasets as well as finetuning on target ("downstream") task or data — both with iterative error backpropagation

B) IDEAL **without finetuning**: the proposed prototype-based IDEAL method involving clustering in the latent feature space with subsequent decision making process such as using winner-takes-all analysis or $k$ nearest neighbours as outlined in Algorithms 1 and 2

C) IDEAL **with finetuning**: Same as B) with the only difference being that the clustering is performed in a latent feature space which is formed by finetuning on target data set (from the "downstream" task) using iterative error backpropagation. Unlike A), setting C) provides interpretable prototypes

The empirical questions are presented below. Questions 1 and 2 confirm that the method delivers competitive results **even without finetuning**. Building upon this initial intuition, we develop the key questions 3, 4 and 5, analysing the performance for lifelong learning scenarios and interpretations proposed by IDEAL, respectively. For reproducibility, the full parameterisation is described in Section A of the Appendix.

**Question 1**. *How does the performance of the IDEAL framework **without** finetuning compare with the well-known deep learning frameworks?*

Section 4.2 and Appendix B show, with a concise summary in Figure 3 and Figure 5, that the gaps between finetuned and non-finetuned IDEAL framework are consistently much smaller (tens of percent vs a few percentage points) for vision transformer backbones comparing to ResNets and VGG. Furthermore, Figure 6 shows that the training time expenditure is more than an order of magnitude smaller comparing to the finetuning time.

**Question 2**. *To what extent does finetuning of the feature space for the target problem lead to overfitting?*

In Section 4.3, figures 8, 9, 10, we demonstrate the issue of overfitting on the target spaces by finetuning on CIFAR-10 and testing on CIFAR-100 in both performance and through visualising the feature space. Interestingly, we also show in Table 3 of the Appendix that, while the choice of prototypes greatly influences the performance of the IDEAL framework **without** finetuning of the backbone, it does not make any significant impact for the finetuned models (i.e., does not improve upon random selection).

**Question 3** *How does the IDEAL framework **without** finetuning compare in the class-incremental learning setting?*

In Section 4.4 we build upon questions 1 and 2 and demonstrate: the small gap between pretrained and finetuned ViT models ultimately enables us to solve class-incremental learning scenarios, improving upon well-known baseline methods. IDEAL framework **without** finetuning shows performance results on a number of class-incremental learning problems, comparable to task-level finetuning. Notably, in CIFAR-100 benchmark, the proposed method provides 83.2% and 69.93% on ViT-L and ResNet-101 respectively, while the state-of-the-art method from (Wang et al. (2022b)) only reports 65.86%.

**Question 4** *How does the IDEAL framework provide insight and interpretation?*

In Section 4.5, we present the analysis of interpretations provided by the method. In Figure 12, 13, 14 we demonstrate the qualitative experiments showing the human-readable interpretations provided by the model for both lifelong learning and offline scenarios.

**Question 5**. *Can models **without** finetuning bring advantage over the finetuned ones in terms of accuracy and help identify misclassifications due to confounding (i.e., spurious correlations in the input)?*

While, admittedly, the model only approaches but does not reach the same level of accuracy for the same backbone without finetuning in the standard benchmarks such as CIFAR-10, it delivers better performance in cases with confounded data (with spurious correlations in the input). In Section 4.6, Table 1 we demonstrate, building upon the intuition from Question 2, that finetuning leads to overfitting on confounded data, and leads to confounded predictions and interpretations. We also demonstrate that in this setting, IDEAL **without finetuning** improves upon F1 score against the finetuned baseline as well as provides interpretations for wrong predictions due to the confounding.

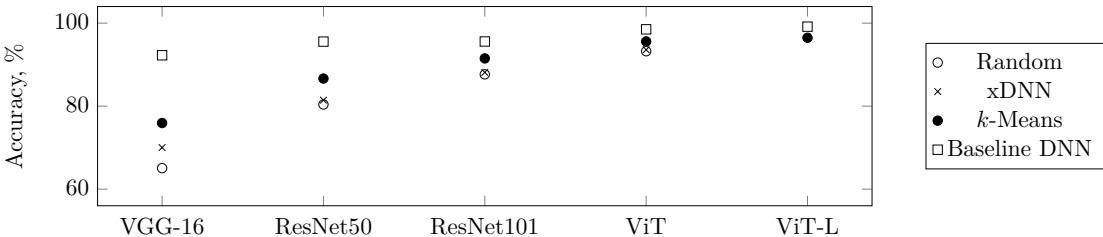

Figure 3: Comparison of the proposed IDEAL framework (**without** finetuning) on the CIFAR-10 data set with different prototype selection methods (random, the clustering used in xDNN (Soares et al. (2021)) and $k$-means method) vs the baseline DNN

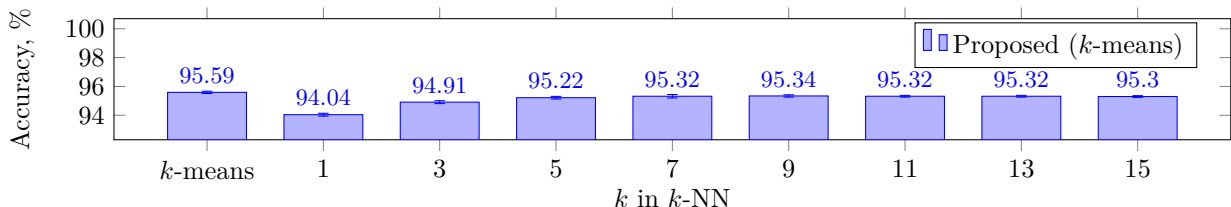

Figure 4: Comparison of results on CIFAR-10 (ViT, $k$ nearest neighbours)

### 4.1 Experimental setting

**Datasets**  CIFAR-10 and CIFAR-100 (Krizhevsky & Hinton (2009)), STL-10 (Coates et al. (2011)), Oxford-IIIT Pet (Parkhi et al. (2012)), EuroSAT (Helber et al. (2018; 2019)), CalTech101 (Li et al. (2006)).

**Feature extractors**  We consider a number of feature extractor networks such as VGG-16 (Simonyan & Zisserman (2014)), RESNET50 (He et al. (2016)), RESNET101 (He et al. (2016)), VIT-B/16 (Dosovitskiy et al. (2020), henceforth referred to as VIT), VIT-L/16 (Dosovitskiy et al. (2020), henceforth referred to as ViT-L) with or **without** finetuning; the pre-trained latent spaces for ViT models were obtained using SWAG methodology (Singh et al. (2022)); the computations for feature extractors has been conducted using a single V100 GPU.

**Prototype selection techniques**  We include the results for such clustering techniques as $k$-means, $k$-means with a nearest data point (referred to as $k$-means (nearest)), and two online clustering methods: xDNN (Angelov & Soares (2020)) and ELM (Baruah & Angelov (2012)).

**Baselines**  We explore trade-offs between standard deep neural networks, different architectural choices (averaged prototypes vs real-world examples) in Section 4.2, and present expanded analysis in Appendix B.

### 4.2 Offline classification

We found that the gap between the finetuned and non-finetuned models on a range of tasks decreases for the modern, high performance, architectures, such as ViT (Dosovitskiy et al. (2020)). For CIFAR-10, these findings are highlighted in Figure 3. While finetuned VGG-16's accuracy is close to the one of ViT and other recent models, different prototype selection techniques **without** finetuning (the one used in xDNN, $k$-means clustering, and random selection) all give accuracy between 60 and 80%. The picture is totally different for ViT, where $k$-means prototype selection **without finetuning** provides accuracy of 95.59% against finetuned ViT's own performance of 98.51%.

While the results above report on performance of the $k$-means clustering used as a prototype selection technique, the experimental results in Figure 4 explore choosing the nearest prototype to $k$-means cluster

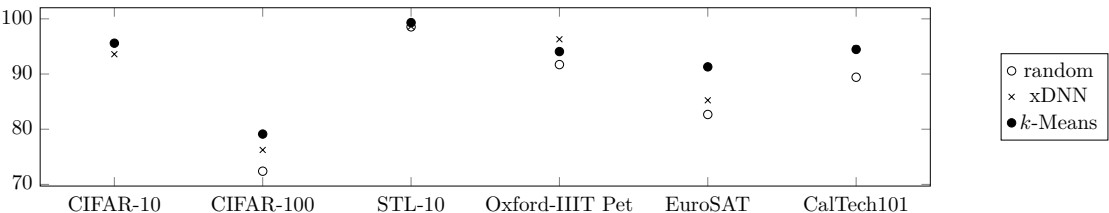

Figure 5: Results **without** finetuning for various problems (ViT)

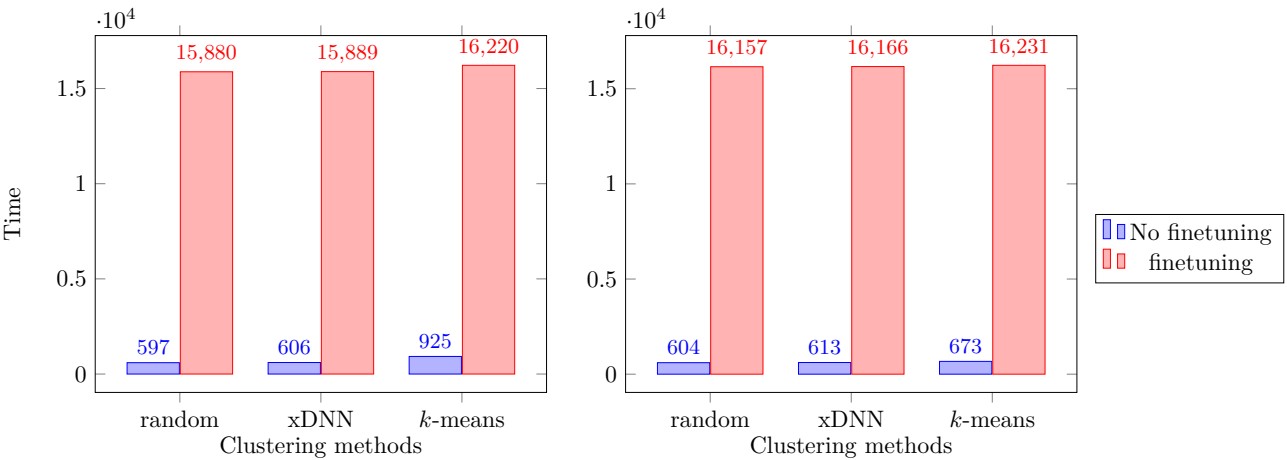

Figure 6: Comparison of training time expenditure on CIFAR-10 (left) and CIFAR-100 (right) with and without funetuning (ViT)

centroid for interpretability reasons. Although it is clear (with further evidence presented in Appendix B) that the performance when selecting the nearest to the $k$-means centroids prototypes is lagging slightly behind the direct use of the centroids (denoted simply as $k$-means), it is possible to bring this performance closer by replacing the winner-takes-all decision making approach (Equation (8)) with the $k$ nearest neighbours method. For this purpose, we utilise the sklearn's `KNeighborsClassifier` function.

The abridged results for classification **without** finetuning for different tasks are presented in Figure 5 (one can find a full version for different methods in Section B).

Below, we analyse closer just the results with using ViT as a feature extractor forming the latent data space. One can see in Figure 7 that: (1) **without finetuning**, on a number of tasks the model shows competitive performance, and (2) with finetuning of the backbone, the difference between the standard backbone and the proposed model is insignificant within the confidence interval. In Figure 6, one can see the comparison of the time expenditure between the finetuned and **non-finetuned** model.

We conducted (see Appendix C) a sensitivity analysis experiment by varying the number of prototypes for CIFAR-10 on ResNet101 backbone by changing the value $k$ for the $k$-means method. In Appendix B, we also show the results with the online clustering method ELM (Baruah & Angelov (2012)), which does not require the number of clusters to be pre-defined and instead uses a radius meta-parameter which affects granulation.

### 4.3 Demonstration of overfitting in the finetuned feature spaces and the prototype selection impact

One clear advantage of transfer learning without finetuning is dramatically lower computational cost reflected in the time expenditure. However, there is also another advantage. The evidence shows that the finetuned feature space shows less generalisation. In Figures 8 and 9, one can see the comparison of the tSNE plots between the finetuned and **non-finetuned** version of the method. While the finetuned method achieves clear

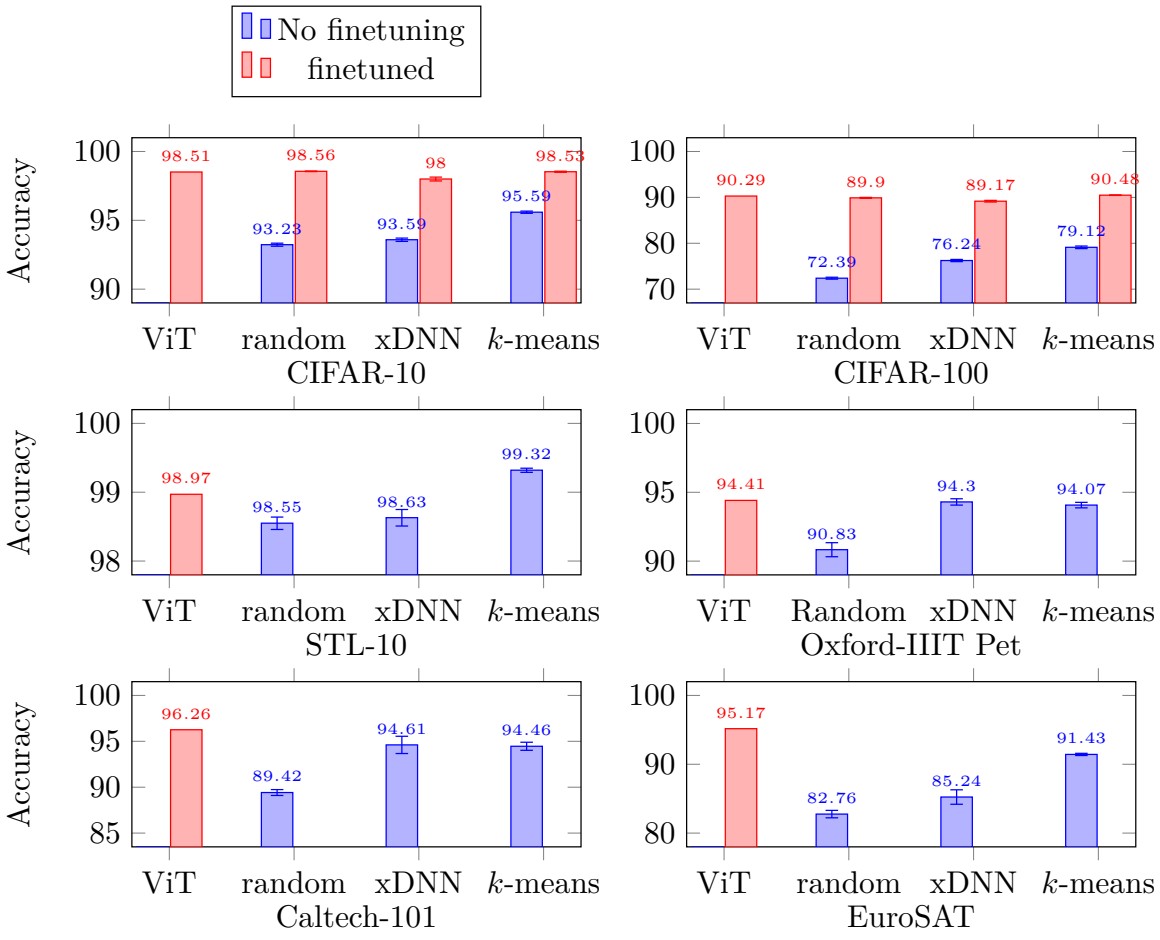

Figure 7: Comparison of results with ViT (Dosovitskiy et al. (2020)) as a feature extractor on a number of datasets. Random, xDNN, $k$-means denote different prototype selection methods

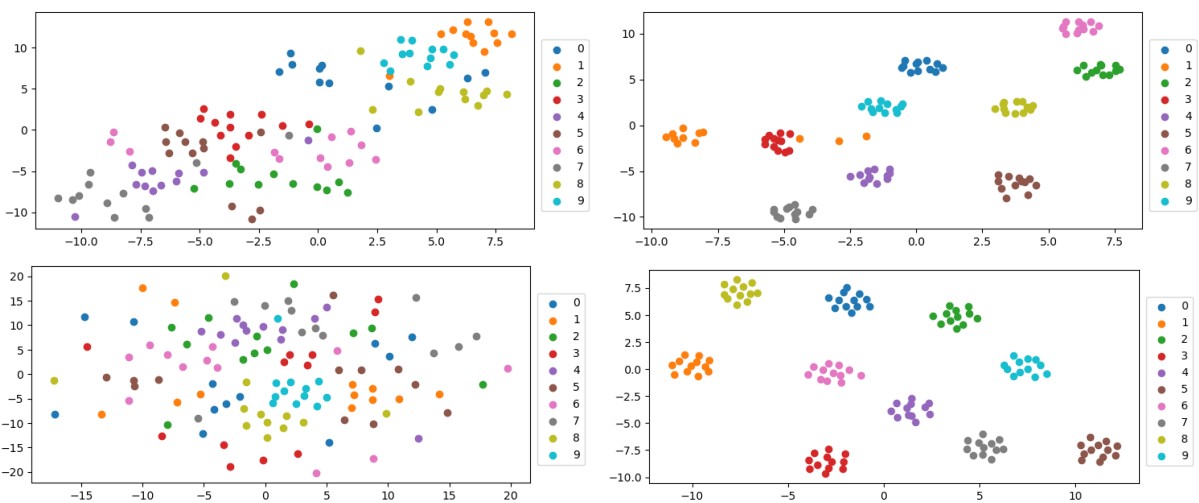

Figure 8: tSNE plots for original (top-left) vs finetuned (top-right) features of ResNet101, k-means prototypes; original (bottom left) vs finetuned (bottom right), ResNet101, random prototype selection, CIFAR-10

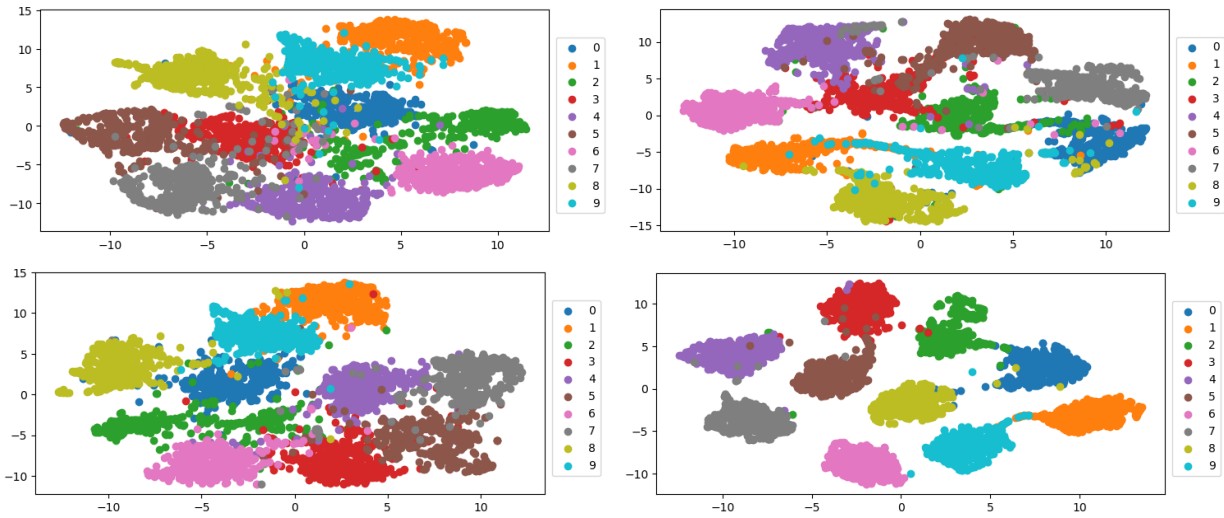

Figure 9: tSNE plots for original (top-left) vs finetuned (top-right) features of ViT, $k$-means prototypes; original (bottom left) vs finetuned (bottom right), ViT, random prototype selection, CIFAR-10

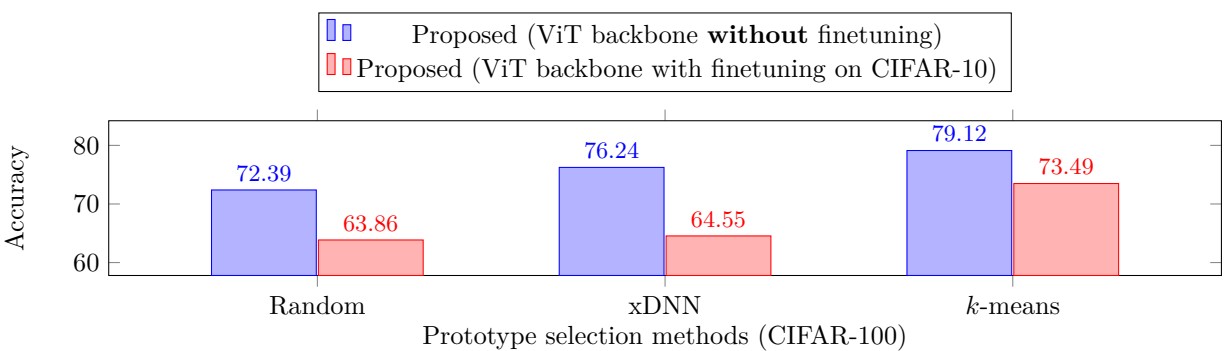

Figure 10: Comparison between the model performance on CIFAR-100 **without** finetuning and finetuning on CIFAR-10 for different prototype selection methods

separation on this task, using the same features to transfer to another task (from CIFAR-10 to CIFAR-100) leads to sharp decrease in performance (see Figure 10).

While for the finetuned backbone, predictably, the results are not far off the standard DL models, they also show no significant difference between different types of prototype selection, including random (see Figure 7). This can be explained by the previous discussion of Figures 8 and 9, which suggests that finetuning gives clear separation of features, so the features of the same class stay close. For the **non-finetuned** results, meanwhile, the difference in accuracy between random and non-random prototype selection is drastic, reaching around 24% for VGG16. This finding remains consistent for a number of vision benchmarks. In Figure 5 and Appendix B, one can see that simple $k$-means prototype selection in the latent space can significantly improve the performance; with the increase of the number of prototypes this difference decreases, but is still present.

## 4.4 Continual learning

The evidence from the previous sections motivates us to extend the analysis to continual learning problems. Given a much smaller gap between the finetuned and non-finetuned ViT models, can the IDEAL framework **without** finetuning compete with the state-of-the-art class-incremental learning baselines? It turns out the answer is affirmative. We repeat the setting from Rebuffi et al. (2017) (Section 4, iCIFAR-100 benchmark) using IDEAL without finetuning the latent space of the ViT-L model. The hyperparameters of the proposed methods are given in Appendix A. This benchmark gradually adds the new classes with a class increment of 10, until it reaches 100 classes. The results, shown in Figure 11a, highlight excellent performance of the proposed method when the number of prototypes is set to 10% of data. As one can see in Appendix C, even much lower number of prototypes, below 1000 or even just 10 per class on average can still lead to competitive results. While we observe $64.18 \pm 0, 0.16, 69.93 \pm 0.23\%, 82.20 \pm 0.23$ for ResNet-50, ResNet-101, and ViT-L respectively, Wang et al. (2022b) reports in its Table 1 for the best performing method for class-incremental learning, based on ViT architecture and contrastive learning, accuracy of just $65.86 \pm 1.24\%$ (with the size of the buffer 1000), while the original benchmark model iCarl (Rebuffi et al. (2017)) reaches, according to Wang et al. (2022b), only $50.49 \pm 0.18\%$.

To demonstrate the consistent performance, we expanded iCIFAR-100 protocol to other datasets, namely class-incremental versions of Caltech101 and CIFAR-10, which we refer to as iCaltech101 and iCIFAR-10. Figure 11 shows robust performance on iCaltech101 and iCIFAR-10. We use the class increment value of ten (eleven for the last step) and two for iCaltech101 and iCIFAR-10, respectively. We see that for iCaltech101, the model performance changes insignificantly when adding the new classes, and all three datasets demonstrate performance similar to offline classification (see Section 4.2).

## 4.5 Study of Interpretability

In Figures 12 and 13, we demonstrate the visual interpretability of the proposed model, through both most similar and most dissimilar prototypes. In addition, the results could be interpreted linguistically (see Appendix D). Figure 13 shows a number of quantitative examples for multiple datasets: Caltech101, STL-10, Oxford-IIIT Pets, all corresponding to the non-finetuned feature space scenario according to the experimental setup from Appendix A. We see that on a range of datasets, without any finetuning, the proposed IDEAL approach provides semantically meaningful interpretations. Furthermore, as there has been no finetuning, the $\ell^2$ distances are defined in exact the same feature space and, hence, can be compared like-for-like between datasets (see subfigures 13a-13f).The experiment in Figure 13 provides an additional reason to use our approach **without** finetuning as it demonstrates that the incorrectly classified data tend to have larger distance to the closest prototypes than the correctly classified ones. Finally, Figure 14 outlines the evolution of predictions for the class-incremental learning scenario. For the sake of demonstration, we used the same setting as the one for the class-incremental lifelong learning detailed in Appendix A and Section 4.4, taking CIFAR-10 for class-incremental learning using ViT model with the increment batch of two classes. We trace the best and the worst matching and select middle prototypes (according to the $\ell^2$ metric) through the stages of class-incremental learning. For the successful predictions, while the best matching prototypes tend to be constant, the worst matching ones change over time when the class changes.

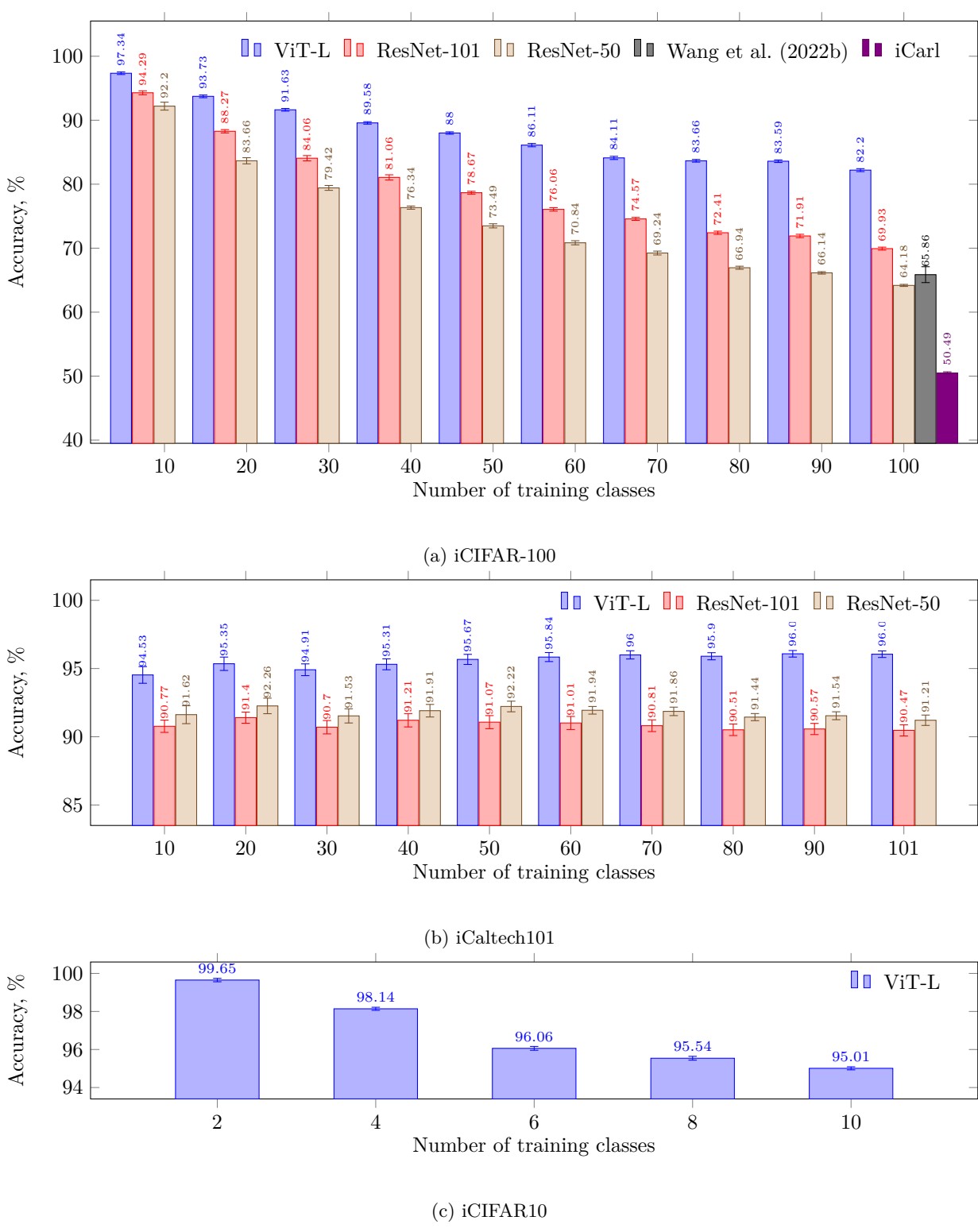

(a) iCIFAR-100

(b) iCaltech101

(c) iCIFAR10

Figure 11: Accuracy of IDEAL in class-incremental learning experiments for different backbones (ViT-L, ResNet-101 and 50); comparison with Wang et al. (2022b) and Rebuffi et al. (2017)

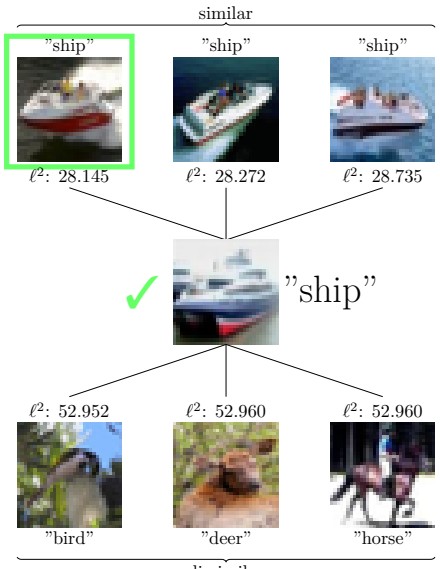

Figure 12: Interpreting the predictions of the proposed model ($k$-means (nearest), CIFAR-10, ViT)

### 4.6  Impact of confounding on interpretations

The phenomenon of confounding takes its origin in causal modelling and is informally described, as per Greenland et al. (1999), as *'a mixing of effects of extraneous factors (called confounders) with the effects of interest'.* In many real-world scenarios, images contain confounding features, such as watermarks or naturally occurring spurious correlations ('seagulls always appear with the sea on the background'). The challenge for the interpretable models is therefore multi-fold: (1) these models need to be resistant to such confounders (2) should these confounders interfere with the performance of the model, the model should highlight them in the interpretations.

To model confounding, we use the experimental setup from Bontempelli et al. (2022), which involves inpainting training images of three out of five selected classes of the CUB dataset with geometric figures (squares) which correlate with, but not caused by, the original data (e.g., every image of the `Crested Auklet` class is marked in the training data with a blue square). In Table 1, we compare the experimental results between the original (Wah et al. (2011)) and confounded (Bontempelli et al. (2022)) CUB dataset. We use the same original pre-trained feature spaces as stated in Appendix A. The finetuned spaces are obtained through finetuning on confounded CUB data from Bontempelli et al. (2022) for 15 epochs.

The results in Table 1 demonstrate clear advantage of models **without finetuning** on the confounded dataset for both $k$-means and $k$-means (nearest), in the case of ViT. Such gap, however, is much narrower for VGG-16 and ResNet-50. It is consistent with the results in Section 4.2 demonstrating the larger finetuning performance advantage for these models compared with the ViT. Furthermore, $k$-means (nearest) does not show improvements over finetuning in a $k$-means (nearest) scenario for VGG-16 and ResNet-50, in a stark contrast with the ViT results.

We demonstrate the interpretations for the confounding experiment in Figure 15. While the model **without** finetuning successfully predicts the correct confounded class, `Black-footed Albatross`, the finetuned model fails at this scenario and predicts a similar class `Sooty Albatross`, which does not contain the confounder mark. On the other hand, the finetuned model performs similarly or better on the original (not confounded) data. These results further build upon the hypothesis from Question 2 and demonstrate that the use of the proposed framework can help address the phenomenon of confounding.

Figure 16 gives an intuition behind improvements in performance of the non-finetuned model. It shows that in the finetuned scenario, confounded training data stands further away from the testing data which does

not contain the confounder mark. In the scenario without finetuning, this does not happen and the training and testing data are matched closer, even in the presence of a confounder. The Sinkhorn approximation of Wasserstein-2 distance has been implemented using `SamplesLoss(loss='sinkhorn', p=2, blur=1e-5)` function from `geomloss` python library.

## 5    Conclusion

Our work shows that interpretable, prototype-based models over the latent spaces of ViT models **without** finetuning, learnt on large generic datasets, work surprisingly well in a number of scenarios. In an extensive set of experiments, we find that:

**Contemporary ViT models drastically narrow the gap between the finetuned and non-finetuned models,** making it possible to avoid finetuning altogether and still have competitive results on a number of benchmarks. To give an example, for VGG-16 backbone, the accuracy difference between the best-performing finetuned and non-finetuned scenarios on CIFAR-10 is 16.61%. The situation is drastically different for the ViT backbone, where this difference is just 2.94%.

**The findings in the previous paragraph indicate that without finetuning we can circumvent the problem of catastrophic forgetting in class-incremental learning.** If the models can achieve competitive performance even **without** finetuning, one can use this advantage to solve a number of problems of lifelong learning without iterative updates and, hence, catastrophic forgetting. The experimental results show the strong empirical advantage of such approach, allowing to achieve, using a ViT-L backbone, a lead of 16.34% on a well-known iCIFAR-100 benchmark.

**The IDEAL framework, proposed in this paper, allows for interpretations** through similarities in the latent feature space, which is not only comparable within one dataset but also between the datasets. We find that the closest prototypes in case of misclassification tend to be further away from the input, and using qualitative analysis, we demonstrate how the IDEAL framework allows to interpret the decision making process in both offline and class-incremental learning scenarios.

**Finetuning results in consistently inferior performance when compared to non-finetuned models in face of confounding bias.** Our initial findings quantify the margin of feature space overfitting in a simple experiment, showing that the ViT model **without** finetuning has 5.63% advantage on CIFAR-100 over the model finetuned on CIFAR-10. We then build upon this observation to show, quantitatively and qualitatively, how the models **without** finetuning outperform the purpose-finetuned counterparts on confounded data. Notably, the model with ViT backbone **without** finetuning achieves 14.1% lead over the finetuned model on confounded CUB dataset with prototypes selected using $k$-means clustering.

## Broader Impact Statement

The IDEAL framework, proposed in this paper, goes beyond the paradigm of first training and then finetuning complex models to the new tasks, which is standard for the field, where both these stages of the approach use expensive GPU compute to improve the model performance. We show that contemporary architectures, trained with extensive datasets, can deliver performance, comparable to task-level finetuning, in a class-incremental learning setting. This can deliver profound impact on democratisation of high-performance machine learning models and implementation on Edge devices, on board of autonomous vehicles, as well as address important problems of environmental sustainability by avoiding using much energy to train and finetune new latent representations, providing instead a way to re-use existing models. Furthermore, the proposed framework can help define a benchmark on how deep-learning latent representations generalise to new tasks.

This approach also naturally extends to task- and potentially, domain-incremental learning, enabling learning new concepts. It demonstrates that with large and complex enough latent spaces, relatively simple strategies of prototype selection, such as clustering, can deliver results comparable with the state-of-the-art in a fraction

| Feature space | Prototype selection | VGG16 | ResNet-50 | ViT |
|---|---|---|---|---|
| Confounded data (Bontempelli et al. (2022)) | | | | |
| Finetuned | N/A, backbone network | $73.99 \pm 2.91$ | $70.42 \pm 2.68$ | $69.06 \pm 4.40$ |
| Non-finetuned | $k$-means | $\mathbf{78.52 \pm 1.31}$ | $\mathbf{76.68 \pm 1.63}$ | $\mathbf{80.70 \pm 2.26}$ |
| Finetuned | $k$-means | $73.19 \pm 1.43$ | $67.16 \pm 2.25$ | $66.58 \pm 5.81$ |
| Non-finetuned | $k$-means (nearest) | $64.13 \pm 1.37$ | $67.68 \pm 0.90$ | $\mathbf{82.88 \pm 2.17}$ |
| Finetuned | $k$-means (nearest) | $\mathbf{71.00 \pm 2.92}$ | $69.03 \pm 1.19$ | $73.99 \pm 5.19$ |
| Original data | | | | |
| Finetuned | N/A, backbone network | $83.66 \pm 1.16$ | $83.49 \pm 1.22$ | $93.92 \pm 1.31$ |
| Non-finetuned | $k$-means | $80.01 \pm 1.27$ | $80.10 \pm 1.66$ | $90.67 \pm 1.13$ |
| Finetuned | $k$-means | $81.98 \pm 1.53$ | $79.38 \pm 2.87$ | $92.85 \pm 1.70$ |
| Non-finetuned | $k$-means (nearest) | $72.11 \pm 1.62$ | $72.64 \pm 1.87$ | $88.57 \pm 0.96$ |
| Finetuned | $k$-means (nearest) | $\mathbf{78.90 \pm 2.77}$ | $\mathbf{80.05 \pm 2.64}$ | $\mathbf{92.80 \pm 1.77}$ |

Table 1: F1 score comparison for CUB dataset (Wah et al. (2011)), %, confidence interval calculated over five runs; all $k$-means runs are for 10% (15) clusters/prototypes; the better results within its category are highlighted in bold, taking into account the confidence interval. While for the original data finetuning has strong performance benefits, non-finetuned model has an edge over the finetuned one for all architectures; for $k$-means (nearest) the non-finetuned model still performs clearly better with ViT architecture than the finetuned counterpart.

of time and compute efforts. Importantly, unlike most of the state-of-the-art approaches, as described in the Related work section of this paper, the proposed framework directly provides interpretability in a linguistic and visual form and provides improved resistance to spurious correlations (confounding bias) in input features.

However, while this study can help advance transparency and trustworthiness of the machine learning models, one needs to duly take into account considerations of privacy and security risks pertinent to deep-learning feature spaces and prototype-based learning. In many cases, such as, notably, for medical applications, there may be a need in preserving privacy of the prototypes and the training data, as exposing prototypes to the users may be unethical or illegal (Lucieri et al. (2023)). Deep-learning models' latent spaces themselves, as well as data they are trained upon, may be biased or unfair (Birhane et al. (2023)). Another risk is a potential for adversarial attacks (Biggio et al. (2013)), affecting either feature space representation or the distances between prototypes.

## Limitations

One of the limitations of this study is that it focuses on the final latent representations and does not analyse the intermediate layers of the common neural network architectures. Future work should consider addressing this issue to improve our understanding of models' inference at a granular level.

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

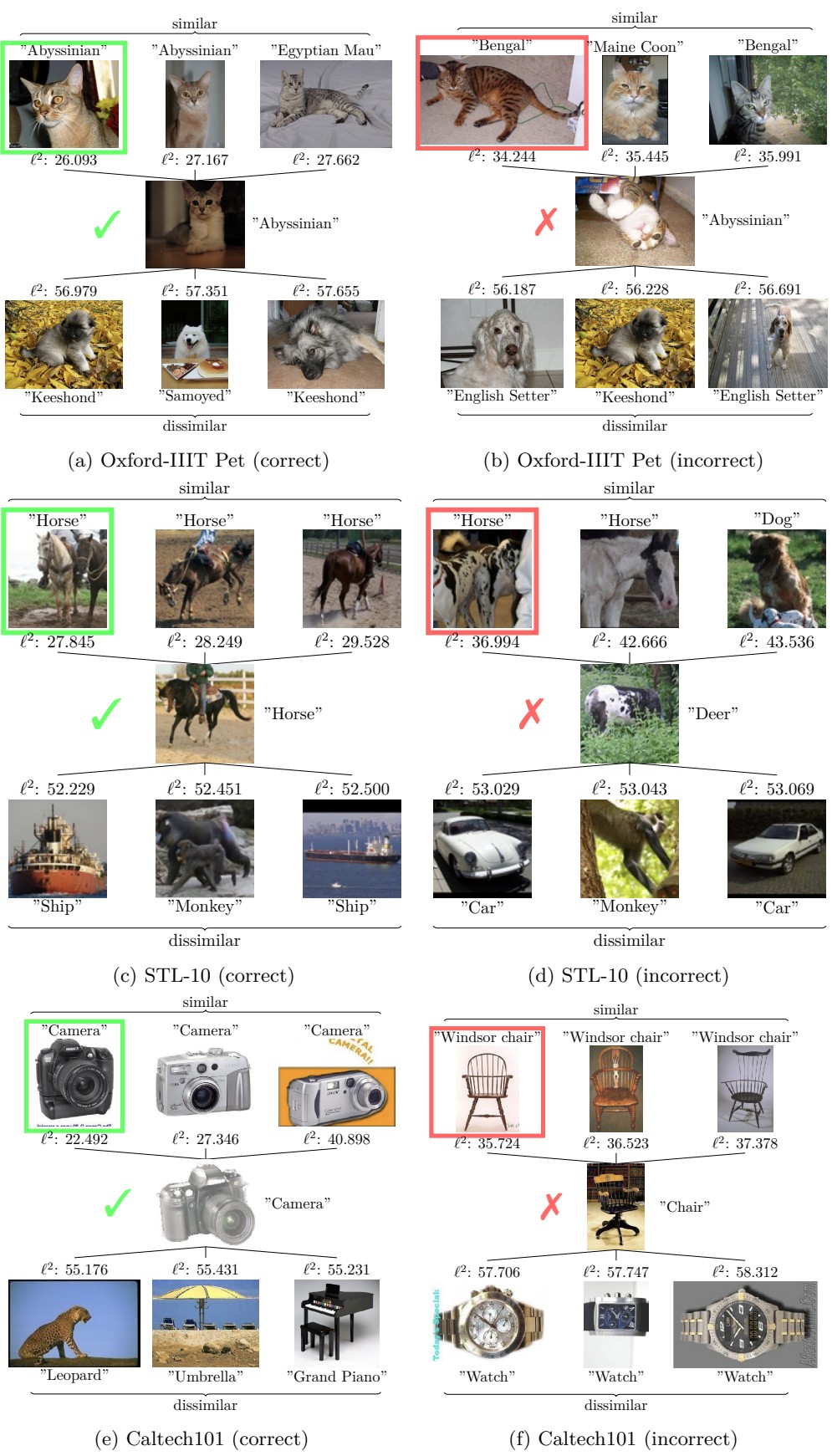

Figure 13: Interpreting the predictions (*k*-means (nearest), OxfordIIITPets/STL-10/Caltech101, ViT)

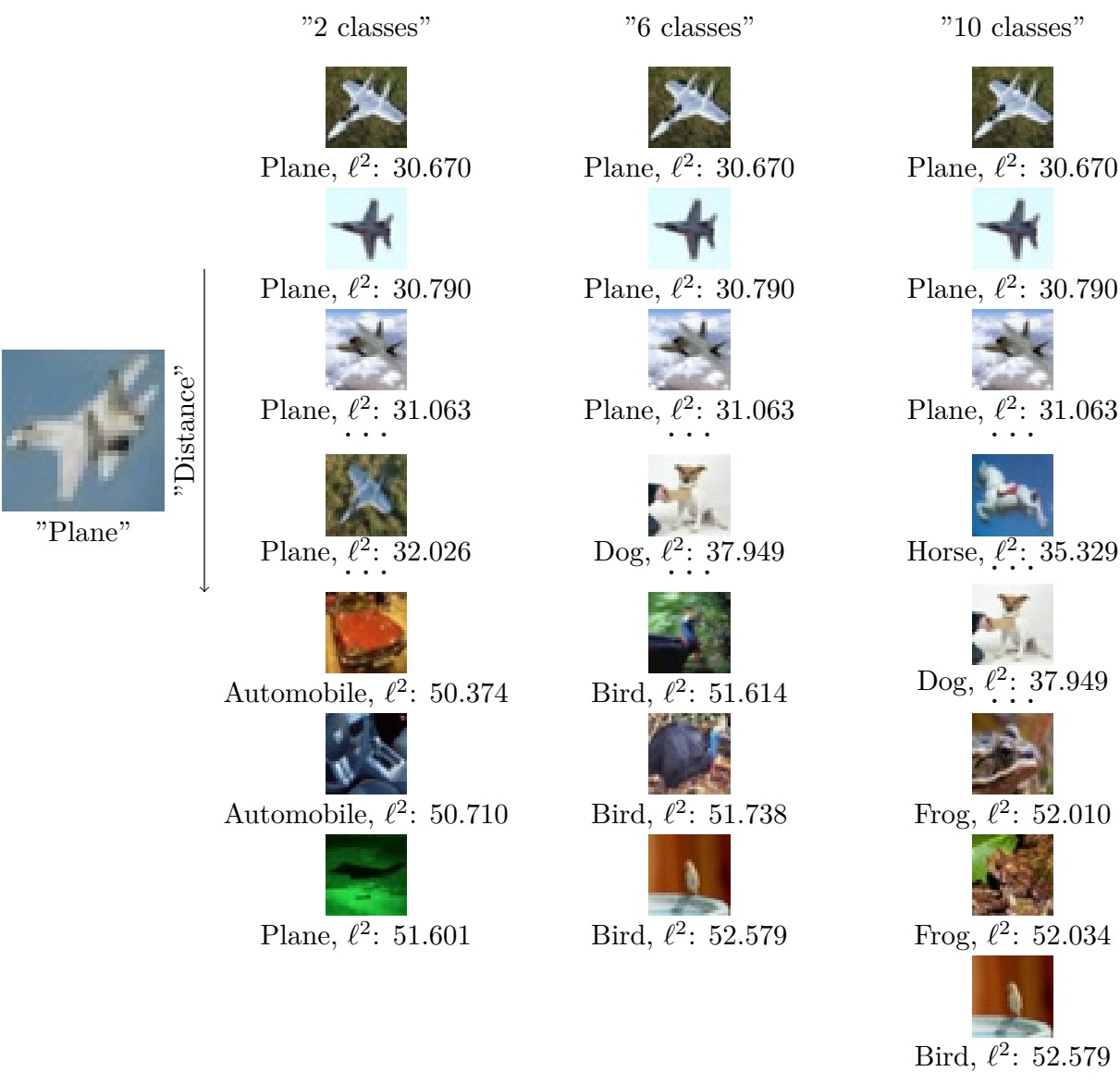

Figure 14: iCIFAR-10 class-incremental learning: evolution of prototype ranking

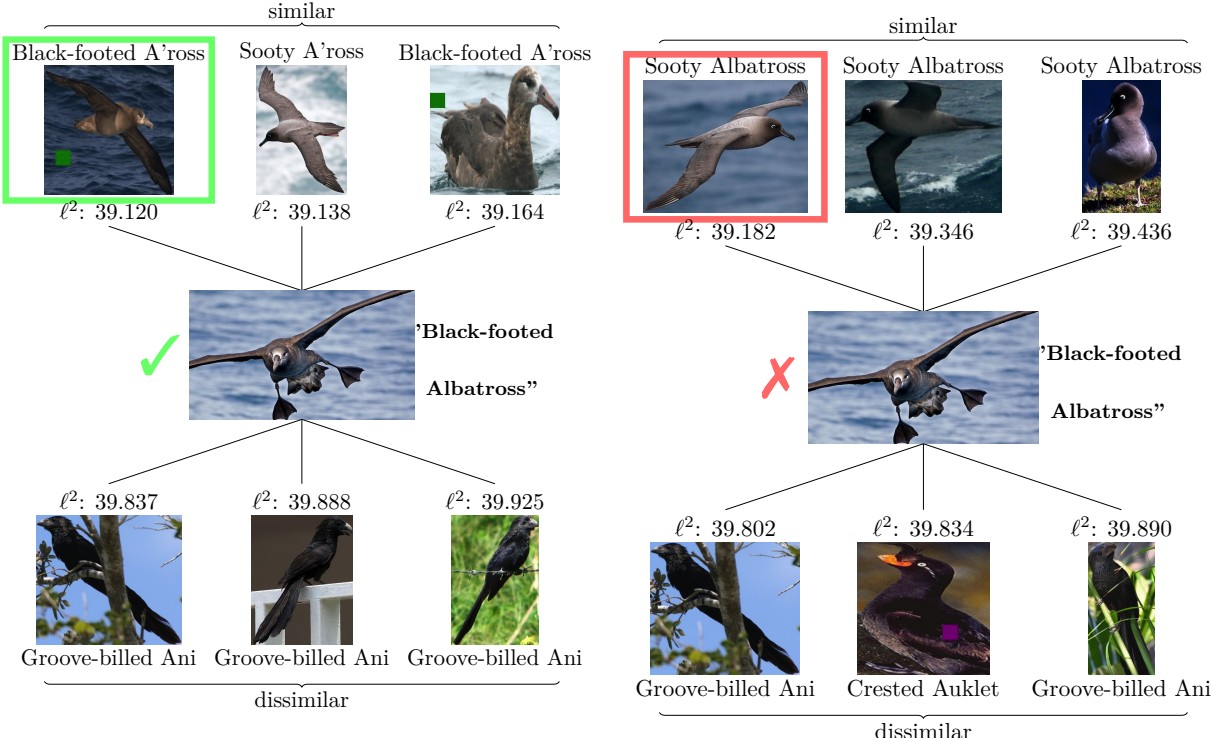

(a) Non-finetuned model interpretation (A'ross denotes 'Albatross')

(b) Finetuned model interpretation

Figure 15: Comparing the interpretations of the non-finetuned and finetuned model with confounding on confounded CUB (Bontempelli et al. (2022)) dataset

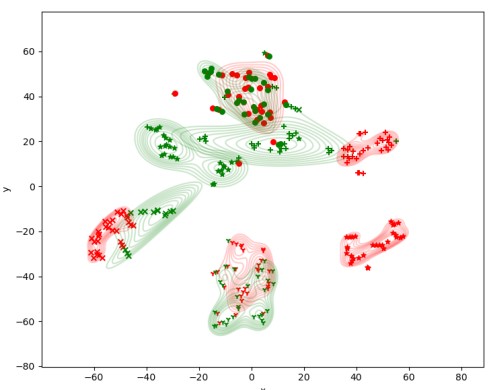 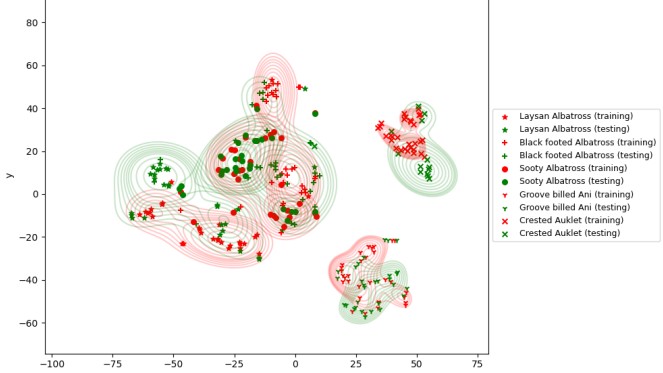

(a) tSNE plot: finetuned model. The clean testing data from confounded classes are better aligned with similar clean classes than with confounded ones

(b) tSNE plot: model without finetuning. The models show better distribution matching, including for similar classes such as different species of Albatross. In both tSNE plots, the density estimation is shown for tSNE embedded points

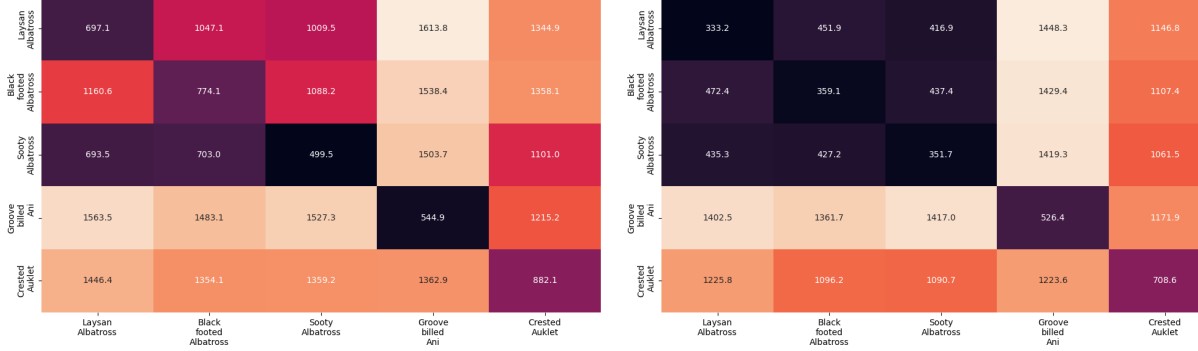

(c) Wasserstein-2 distance heatmap (Sinkhorn approximation): finetuned model, training (vertical) to testing (horizontal) distance. `Black-footed albatross` (testing distribution) is closer to a non-confounded `Sooty albatross` training distribution

(d) Wasserstein-2 distance heatmap (Sinkhorn approximation): model without finetuning. In contrast to the finetuned model, the similar classes' distributions are close yet closely match between training and testing classes.

Figure 16: Intuitive explanation behind better performance of non-finetuned model

Plamen Angelov and Eduardo Soares. Towards explainable deep neural networks (xdnn). *Neural Networks*, 130:185–194, 2020.

Plamen Angelov and Xiaowei Zhou. Evolving fuzzy-rule-based classifiers from data streams. *Ieee transactions on fuzzy systems*, 16(6):1462–1475, 2008.

Nachman Aronszajn. Theory of reproducing kernels. *Transactions of the American mathematical society*, 68(3):337–404, 1950.

Akanksha Atrey, Kaleigh Clary, and David Jensen. Exploratory not explanatory: Counterfactual analysis of saliency maps for deep reinforcement learning. *arXiv preprint arXiv:1912.05743*, 2019.

Rashmi Dutta Baruah and Plamen Angelov. Evolving local means method for clustering of streaming data. In *2012 IEEE international conference on fuzzy systems*, pp. 1–8. IEEE, 2012.

Dimitris Bertsimas, Angela King, and Rahul Mazumder. Best subset selection via a modern optimisation lens. *The Annals of Statistics*, 44(2):813–852, 2016.

Jacob Bien and Robert Tibshirani. Prototype selection for interpretable classification. *The Annals of Applied Statistics*, pp. 2403–2424, 2011.

Battista Biggio, Igino Corona, Davide Maiorca, Blaine Nelson, Nedim Šrndić, Pavel Laskov, Giorgio Giacinto, and Fabio Roli. Evasion attacks against machine learning at test time. In *Machine Learning and Knowledge Discovery in Databases: European Conference, ECML PKDD 2013, Prague, Czech Republic, September 23-27, 2013, Proceedings, Part III 13*, pp. 387–402. Springer, 2013.

Abeba Birhane, Vinay Prabhu, Sang Han, Vishnu Naresh Boddeti, and Alexandra Sasha Luccioni. Into the laions den: Investigating hate in multimodal datasets. *arXiv preprint arXiv:2311.03449*, 2023.

Moritz Böhle, Mario Fritz, and Bernt Schiele. B-cos networks: alignment is all we need for interpretability. In *Proceedings of the IEEE/CVF Conference on Computer Vision and Pattern Recognition*, pp. 10329–10338, 2022.

Andrea Bontempelli, Stefano Teso, Katya Tentori, Fausto Giunchiglia, and Andrea Passerini. Concept-level debugging of part-prototype networks. *arXiv preprint arXiv:2205.15769*, 2022.

Bernhard Boser, Isabelle Guyon, and Vladimir Vapnik. A training algorithm for optimal margin classifiers. In *Proceedings of the fifth annual workshop on Computational learning theory*, pp. 144–152, 1992.

Chaofan Chen, Oscar Li, Daniel Tao, Alina Barnett, Cynthia Rudin, and Jonathan K Su. This looks like that: deep learning for interpretable image recognition. *Advances in neural information processing systems*, 32, 2019.

Adam Coates, Andrew Ng, and Honglak Lee. An analysis of single-layer networks in unsupervised feature learning. In *Proceedings of the fourteenth international conference on artificial intelligence and statistics*, pp. 215–223. JMLR Workshop and Conference Proceedings, 2011.

Dorin Comaniciu and Peter Meer. Mean shift: A robust approach toward feature space analysis. *IEEE Transactions on pattern analysis and machine intelligence*, 24(5):603–619, 2002.

Marco Cuturi. Sinkhorn distances: Lightspeed computation of optimal transport. *Advances in neural information processing systems*, 26, 2013.

Alexey Dosovitskiy, Lucas Beyer, Alexander Kolesnikov, Dirk Weissenborn, Xiaohua Zhai, Thomas Unterthiner, Mostafa Dehghani, Matthias Minderer, Georg Heigold, Sylvain Gelly, et al. An image is worth 16x16 words: Transformers for image recognition at scale. *arXiv preprint arXiv:2010.11929*, 2020.

Robert French. Catastrophic forgetting in connectionist networks. *Trends in cognitive sciences*, 3(4):128–135, 1999.

Sander Greenland, Judea Pearl, and James M Robins. Confounding and collapsibility in causal inference. *Statistical science*, 14(1):29–46, 1999.

Kaiming He, Xiangyu Zhang, Shaoqing Ren, and Jian Sun. Deep residual learning for image recognition. In *Proceedings of the IEEE conference on computer vision and pattern recognition*, pp. 770–778, 2016.

Patrick Helber, Benjamin Bischke, Andreas Dengel, and Damian Borth. Introducing eurosat: A novel dataset and deep learning benchmark for land use and land cover classification. In *IGARSS 2018-2018 IEEE International Geoscience and Remote Sensing Symposium*, pp. 204–207. IEEE, 2018.

Patrick Helber, Benjamin Bischke, Andreas Dengel, and Damian Borth. Eurosat: A novel dataset and deep learning benchmark for land use and land cover classification. *IEEE Journal of Selected Topics in Applied Earth Observations and Remote Sensing*, 2019.

Been Kim, Cynthia Rudin, and Julie A Shah. The bayesian case model: A generative approach for case-based reasoning and prototype classification. *Advances in neural information processing systems*, 27, 2014.

Jinkyu Kim and John Canny. Interpretable learning for self-driving cars by visualizing causal attention. In *Proceedings of the IEEE international conference on computer vision*, pp. 2942–2950, 2017.

James Kirkpatrick, Razvan Pascanu, Neil Rabinowitz, Joel Veness, Guillaume Desjardins, Andrei A Rusu, Kieran Milan, John Quan, Tiago Ramalho, Agnieszka Grabska-Barwinska, et al. Overcoming catastrophic forgetting in neural networks. *Proceedings of the national academy of sciences*, 114(13):3521–3526, 2017.

Simon Kornblith, Jonathon Shlens, and Quoc V Le. Do better imagenet models transfer better? In *Proceedings of the IEEE/CVF conference on computer vision and pattern recognition*, pp. 2661–2671, 2019.

Alex Krizhevsky and Geoffrey Hinton. Learning multiple layers of features from tiny images. Technical Report 0, University of Toronto, Toronto, Ontario, 2009.

Alex Krizhevsky, Ilya Sutskever, and Geoffrey E Hinton. Imagenet classification with deep convolutional neural networks. *Advances in Neural Information Processing Systems*, 25, 2012.

Christiaan Lamers, René Vidal, Nabil Belbachir, Niki van Stein, Thomas Bäeck, and Paris Giampouras. Clustering-based domain-incremental learning. In *Proceedings of the IEEE/CVF International Conference on Computer Vision*, pp. 3384–3392, 2023.

Fei-Fei Li, Rob Fergus, and Pietro Perona. One-shot learning of object categories. *IEEE transactions on pattern analysis and machine intelligence*, 28(4):594–611, 2006.

Oscar Li, Hao Liu, Chaofan Chen, and Cynthia Rudin. Deep learning for case-based reasoning through prototypes: A neural network that explains its predictions. In *Proceedings of the AAAI Conference on Artificial Intelligence*, volume 32, 2018.

Zhizhong Li and Derek Hoiem. Learning without forgetting. *IEEE transactions on pattern analysis and machine intelligence*, 40(12):2935–2947, 2017.

Adriano Lucieri, Andreas Dengel, and Sheraz Ahmed. Translating theory into practice: assessing the privacy implications of concept-based explanations for biomedical ai. *Frontiers in Bioinformatics*, 3, 2023.

James MacQueen et al. Some methods for classification and analysis of multivariate observations. In *Proceedings of the fifth Berkeley symposium on mathematical statistics and probability*, volume 1, pp. 281–297. Oakland, CA, USA, 1967.

Balas Kausik Natarajan. Sparse approximate solutions to linear systems. *SIAM journal on computing*, 24 (2):227–234, 1995.

Meike Nauta, Ron Van Bree, and Christin Seifert. Neural prototype trees for interpretable fine-grained image recognition. In *Proceedings of the IEEE/CVF Conference on Computer Vision and Pattern Recognition*, pp. 14933–14943, 2021.

Allen Newell, John C Shaw, and Herbert A Simon. Report on a general problem solving program. In *IFIP congress*, volume 256, pp. 64. Pittsburgh, PA, 1959.

Maxime Oquab, Timothée Darcet, Théo Moutakanni, Huy Vo, Marc Szafraniec, Vasil Khalidov, Pierre Fernandez, Daniel Haziza, Francisco Massa, Alaaeldin El-Nouby, et al. Dinov2: Learning robust visual features without supervision. *arXiv preprint arXiv:2304.07193*, 2023.

German Parisi, Ronald Kemker, Jose L Part, Christopher Kanan, and Stefan Wermter. Continual lifelong learning with neural networks: A review. *Neural networks*, 113:54–71, 2019.

Omkar Parkhi, Andrea Vedaldi, Andrew Zisserman, and CV Jawahar. Cats and dogs. In *2012 IEEE conference on computer vision and pattern recognition*, pp. 3498–3505. IEEE, 2012.

Uwe Peters. Explainable ai lacks regulative reasons: why ai and human decision-making are not equally opaque. *AI and Ethics*, pp. 1–12, 2022.

Tomaso Poggio and Federico Girosi. A sparse representation for function approximation. *Neural computation*, 10(6):1445–1454, 1998.

Milos Radovanovic, Alexandros Nanopoulos, and Mirjana Ivanovic. Hubs in space: Popular nearest neighbors in high-dimensional data. *Journal of Machine Learning Research*, 11(sept):2487–2531, 2010.

Sylvestre-Alvise Rebuffi, Alexander Kolesnikov, Georg Sperl, and Christoph Lampert. icarl: Incremental classifier and representation learning. In *2017 IEEE Conference on Computer Vision and Pattern Recognition (CVPR)*, pp. 5533–5542. IEEE, 2017.

Frank Rosenblatt et al. *Principles of neurodynamics: Perceptrons and the theory of brain mechanisms*, volume 55. Spartan books Washington, DC, 1962.

Cynthia Rudin. Stop explaining black box machine learning models for high stakes decisions and use interpretable models instead. *Nature Machine Intelligence*, 1(5):206–215, 2019. doi: 10.1038/s42256-019-0048-x. URL https://doi.org/10.1038/s42256-019-0048-x.

David E Rumelhart, Geoffrey E Hinton, and Ronald J Williams. Learning representations by back-propagating errors. *nature*, 323(6088):533–536, 1986.

Paul Ruvolo and Eric Eaton. Ella: An efficient lifelong learning algorithm. In *International conference on machine learning*, pp. 507–515. PMLR, 2013.

Bernhard Schölkopf, Ralf Herbrich, and Alex J Smola. A generalized representer theorem. In *Computational Learning Theory: 14th Annual Conference on Computational Learning Theory, COLT 2001 and 5th European Conference on Computational Learning Theory, EuroCOLT 2001 Amsterdam, The Netherlands, July 16–19, 2001 Proceedings 14*, pp. 416–426. Springer, 2001.

Ramprasaath R Selvaraju, Michael Cogswell, Abhishek Das, Ramakrishna Vedantam, Devi Parikh, and Dhruv Batra. Grad-cam: Visual explanations from deep networks via gradient-based localization. In *Proceedings of the IEEE international conference on computer vision*, pp. 618–626, 2017.

Karen Simonyan and Andrew Zisserman. Very deep convolutional networks for large-scale image recognition. *arXiv preprint arXiv:1409.1556*, 2014.

Karen Simonyan, Andrea Vedaldi, and Andrew Zisserman. Deep inside convolutional networks: Visualising image classification models and saliency maps. 2014.

Mannat Singh, Laura Gustafson, Aaron Adcock, Vinicius de Freitas Reis, Bugra Gedik, Raj Prateek Kosaraju, Dhruv Mahajan, Ross Girshick, Piotr Dollár, and Laurens van der Maaten. Revisiting Weakly Supervised Pre-Training of Visual Perception Models. In *CVPR*, 2022.

Alex Smola and Bernhard Schölkopf. A tutorial on support vector regression. *Statistics and computing*, 14:199–222, 2004.

Jake Snell, Kevin Swersky, and Richard Zemel. Prototypical networks for few-shot learning. *Advances in neural information processing systems*, 30, 2017.

Eduardo Soares, Plamen Angelov, and Ziyang Zhang. An explainable approach to deep learning from ct-scans for covid identification. 2021.

Hugo Steinhaus et al. Sur la division des corps matériels en parties. *Bull. Acad. Polon. Sci*, 1(804):801, 1956.

Michael Tipping. The relevance vector machine. *Advances in neural information processing systems*, 12, 1999.

Michael Tipping. Sparse bayesian learning and the relevance vector machine. *Journal of machine learning research*, 1(Jun):211–244, 2001.

Gido van de Ven, Tinne Tuytelaars, and Andreas S Tolias. Three types of incremental learning. *Nature Machine Intelligence*, pp. 1–13, 2022.

Catherine Wah, Steve Branson, Peter Welinder, Pietro Perona, and Serge Belongie. The caltech-ucsd birds-200-2011 dataset. 2011.

Wenguan Wang, Cheng Han, Tianfei Zhou, and Dongfang Liu. Visual recognition with deep nearest centroids. In *International Conference on Learning Representations (ICLR)*, 2023.

Yabin Wang, Zhiwu Huang, and Xiaopeng Hong. S-prompts learning with pre-trained transformers: An occam's razor for domain incremental learning. *Advances in Neural Information Processing Systems*, 35: 5682–5695, 2022a.

Zhen Wang, Liu Liu, Yajing Kong, Jiaxian Guo, and Dacheng Tao. Online continual learning with contrastive vision transformer. In *Computer Vision–ECCV 2022: 17th European Conference, Tel Aviv, Israel, October 23–27, 2022, Proceedings, Part XX*, pp. 631–650. Springer, 2022b.

Dennis Wei, Rahul Nair, Amit Dhurandhar, Kush R Varshney, Elizabeth Daly, and Moninder Singh. On the safety of interpretable machine learning: A maximum deviation approach. *Advances in Neural Information Processing Systems*, 35:9866–9880, 2022.

Gerhard Widmer and Miroslav Kubat. Learning in the presence of concept drift and hidden contexts. *Machine learning*, 23:69–101, 1996.

Shipeng Yan, Jiangwei Xie, and Xuming He. Der: Dynamically expandable representation for class incremental learning. In *Proceedings of the IEEE/CVF Conference on Computer Vision and Pattern Recognition*, pp. 3014–3023, 2021.

Dagmar Zeithamova, W Todd Maddox, and David M Schnyer. Dissociable prototype learning systems: evidence from brain imaging and behavior. *Journal of Neuroscience*, 28(49):13194–13201, 2008.

Ziyang Zhang, Plamen Angelov, Eduardo Soares, Nicolas Longepe, and Pierre Philippe Mathieu. An interpretable deep semantic segmentation method for earth observation. *arXiv preprint arXiv:2210.12820*, 2022.

Junxian Zhu, Canhong Wen, Jin Zhu, Heping Zhang, and Xueqin Wang. A polynomial algorithm for best-subset selection problem. *Proceedings of the National Academy of Sciences*, 117(52):33117–33123, 2020.

## A  Experimental setup

In this work, all the experiments were conducted in PyTorch 2.0.0. The pre-trained models used in these experiments were obtained from TorchVision [1] while the finetuned models have been obtained from three different sources:

1. *Models that come from MMPreTrain* [2]. Specifically, ResNet50 and ResNet101 finetuned on the CIFAR-10, and ResNet 50 finetuned on CIFAR-100.

2. *finetuned TorchVision models.* finetuning was conducted by continuing the EBP across all network layers. Such models include VGG-16 and Vision Transformer (ViT) finetuned on CIFAR-10, as well as ResNet101, VGG-16, and ViT finetuned on CIFAR-100. For ResNet101 and VGG-16 models, we ran the training for 200 epochs, while the Vision Transformer models were trained for 10 epochs. The Stochastic Gradient Descent (SGD) optimizer was employed for all models, with a learning rate of 0.0005 and a momentum value of 0.9.

3. *Linearly finetuned TorchVision models.* In such case, only the linear classifier was trained and all the remaining layers of the network were fixed. For these models, we conducted training for 200 epochs for ResNet50, ResNet101, and VGG16, and 25 epochs for the ViT models. We adopted the Stochastic Gradient Descent (SGD) optimizer, with a learning rate of 0.001 and a momentum parameter set at 0.9.

We utilized $k$-means clustering and random selection methods, setting the number of prototypes for each class at 10% of the training data for the corresponding classes. Besides, we also set it to 12 per class and conducted experiments for ResNet50, ResNet101, and VGG-16 on CIFAR-10 and CIFAR-100 datasets, enabling us to evaluate the impact of varying the number of prototypes.

For ELM online clustering method, we experimented with varying radius values for each specific dataset and backbone network. We selected a radius value that would maintain the number of prototypes within the range of 0-20% of the training data. In the experiments without finetuning on the CIFAR-10 dataset, we set the radius to 8, 10, 19, and 12 for ResNet50, ResNet101, VGG-16, and Vision Transformer (ViT) models respectively. The radius was adjusted to 8, 11, 19, and 12 for these models when conducting the same tasks without finetuning on CIFAR-100. For STL10, Oxford-IIIT Pets, EuroSAT, and CalTech101 datasets, the radius was set to 13 across all ELM experiments. In contrast, the xDNN model did not require hyper-parameter settings as it is inherently a parameter-free model.

We performed all experiments for Sections 4.2 and 4.4 of the main paper 5 times and report mean values and standard deviations for our results, with the exception of the finetuned backbone models where we just performed finetuning once (or sourced finetuned models as detailed above).

For the class-incremental learning experiments in Section 4.4, we use the $k$-means clustering method for prototype selection and set the number of prototypes to 10% of the training data. Each time we add the incremental classes, the existing prototypes are unchanged, and the algorithm adds the prototypes for the new classes to the existing prototypes. All the experiments were executed 10 times to allow a robust comparison with the benchmark results.

To ensure a consistent and stable training environment, for every experiment we used a single NVIDIA V100 GPU from a cluster.

## B  Complete experimental results

Tables 2-9 contain extended experimental results for multiple benchmarks and feature extractors. These results further support the findings of the main paper.

---

[1] https://pytorch.org/vision/main/models.html
[2] https://github.com/open-mmlab/mmpretrain

| FE | method | accuracy (%) | #prototypes | time, s |
|---|---|---|---|---|
| ResNet50 | random | $65.55 \pm 1.93$ | $120(0.24\%)$ | 85 |
| | random | $80.40 \pm 0.37$ | $5,000(10\%)$ | 85 |
| | ELM | $81.17 \pm 0.04$ | $5,500(11\%)$ | 365 |
| | xDNN | $81.44 \pm 0.33$ | $115(0.23\%)$ | 103 |
| | $k$-means | $84.12 \pm 0.19$ | $120(0.24\%)$ | 201 |
| | $k$-means | $\mathbf{86.65 \pm 0.15}$ | $5,000(10\%)$ | $1,138$ |
| ResNet101 | random | $78.08 \pm 1.38$ | $120(0.24\%)$ | 129 |
| | random | $87.66 \pm 0.25$ | $5,000(10\%)$ | 129 |
| | ELM | $88.22 \pm 0.09$ | $7,154(14.31\%)$ | 524 |
| | xDNN | $88.13 \pm 0.42$ | $118(0.24\%)$ | 145 |
| | $k$-means | $90.19 \pm 0.15$ | $120(0.24\%)$ | 245 |
| | $k$-means | $\mathbf{91.50 \pm 0.07}$ | $5,000(10\%)$ | $1,194$ |
| VGG-16 | random | $50.13 \pm 2.37$ | $120(0.24\%)$ | 95 |
| | random | $65.06 \pm 0.32$ | $5,000(10\%)$ | 95 |
| | ELM | $72.31 \pm 0.08$ | $1,762(3.52\%)$ | 215 |
| | xDNN | $70.03 \pm 0.96$ | $103(0.21\%)$ | 132 |
| | $k$-means | $74.48 \pm 0.16$ | $120(0.24\%$ | 346 |
| | $k$-means | $\mathbf{75.94 \pm 0.15}$ | $5,000(10\%)$ | $2,362$ |
| ViT | random | $93.23 \pm 0.11$ | $5,000(10\%)$ | 597 |
| | ELM | $90.61 \pm 0.14$ | $6,685(13.37\%)$ | 889 |
| | xDNN | $93.59 \pm 0.12$ | $112(0.2\%)$ | 606 |
| | $k$-means | $\mathbf{95.59 \pm 0.08}$ | $5,000(10\%)$ | 925 |
| ViT-L | $k$-means | $\mathbf{96.48 \pm 0.05}$ | $5,000(10\%)$ | $4,375$ |
| | $k$-means (nearest) | $95.62 \pm 0.07$ | $5,000(10\%)$ | $4,352$ |

Table 2: CIFAR-10 classification task comparison for the case of no finetuning of the feature extractor

Table 2 demonstrates the data behind Figure 3 of the main paper. It also highlights the performance of the $k$-means model on ViT-L latent space, when the nearest real training data point to the $k$-means cluster centre is selected (labelled as $k$-means (nearest)). One can also see that even with the small number of selected prototypes, the algorithm delivers competitive performance without finetuning.

Table 3 compares different latent spaces and gives the number of free (optimised) parameters for the scenario of finetuning of the models. With a small additional number of parameters, which is the number of possible prototypes, one can transform the opaque architectures into ones interpretable through proximity and similarity to prototypes within the latent space (this is highlighted in the interpretability column).

Tables 4-9 repeat the same analysis, expanded from Figure 5 of the main paper for different datasets. The results show remarkable consistency with the previous conclusions and further back up the claims of generalisation to different classification tasks.

## C Sensitivity analysis for the number of prototypes

Figure 17 further backs up the previous evidence that even with a small number of prototypes, the accuracy is still high. It shows, however, that there is a trade-off between the number of prototypes and accuracy. It also shows, that after a few hundred prototypes per class on CIFAR-10 and CIFAR-100 tasks, the performance does not increase and may even slightly decrease, indicating saturation.

## D Linguistic interpretability of the proposed framework outputs

To back up interpretability claim, we present two additional interpretability scenarios complementing the one in Section 4.5 of the main text.

| FE | method | accuracy (%) | #parameters | #prototypes | time, s | interpretability |
|---|---|---|---|---|---|---|
| ResNet50 | ResNet50 | 95.55 (80.71*) | $\sim 25M$ (20K) | | 36,360 (13,122*) | ✗ |
| | random | 94.92 ± 0.02 | $\sim 25M + 50K$ | 120(0.24%) | 36,360 + 24 | ✓ |
| | random | 95.32 ± 0.09 | $\sim 25M + 50K$ | 5,000(10%) | 36,360 + 24 | ✓ |
| | xDNN | 95.32 ± 0.12 | $\sim 25M + 50K$ | 111(0.22%) | 36,360 + 43 | ✓ |
| | $k$-means | 94.91 ± 0.14 | $\sim 25M + 50K$ | 120(0.24%) | 36,360 + 208 | ✓ |
| | $k$-means | 95.50 ± 0.06 | $\sim 25M + 50K$ | 5,000(10%) | 36,360 + 1,288 | ✓ |
| ResNet101 | Resnet101 | 95.58 (84.44*) | $\sim 44M$ (20K) | | 36,360 | ✗ |
| | random | 95.47 ± 0.06 | $\sim 44M + 50K$ | 120(0.24%) | 36,360 + 37 | ✓ |
| | random | 95.51 ± 0.01 | $\sim 44M + 50K$ | 5,000(10%) | 36,360 + 37 | ✓ |
| | xDNN | 95.50 ± 0.10 | $\sim 44M + 50K$ | 107(0.21%) | 36,360 + 54 | ✓ |
| | $k$-means | 95.55 ± 0.03 | $\sim 44M + 50K$ | 120(0.24%) | 36,360 + 231 | ✓ |
| | $k$-means | 95.51 ± 0.04 | $\sim 44M + 50K$ | 5,000(10%) | 36,360 + 1,357 | ✓ |
| VGG-16 | VGG-16 | 92.26 (83.71*) | $\sim 138M$ (41K) | | 40,810 | ✗ |
| | random | 87.48 ± 0.72 | $\sim 138M + 50K$ | 120(0.24%) | 40,810 + 94 | ✓ |
| | random | 90.86 ± 0.19 | $\sim 138M + 50K$ | 5,000(10%) | 40,810 + 94 | ✓ |
| | xDNN | 91.42 ± 0.25 | $\sim 138M + 50K$ | 102(0.20%) | 40,810 + 123 | ✓ |
| | $k$-means | 92.24 ± 0.10 | $\sim 138M + 50K$ | 120(0.24%) | 40,810 + 369 | ✓ |
| | $k$-means | 92.55 ± 0.16 | $\sim 138M + 50K$ | 5,000(10%) | 40,810 + 2,408 | ✓ |
| ViT | ViT | 98.51 (96.08*) | $\sim 86M$ (8K) | | 15,282 (15,565*) | ✗ |
| | random | 98.56 ± 0.02 | $\sim 86M + 50K$ | 5,000(10%) | 15,282 + 598 | ✓ |
| | xDNN | 98.00 ± 0.14 | $\sim 86M + 50K$ | 117(0.23%) | 15,282 + 607 | ✓ |
| | $k$-means | 98.53 ± 0.04 | $\sim 86M + 50K$ | 5,000(10%) | 15,282 + 938 | ✓ |

Table 3: CIFAR-10 classification task comparison for the case of finetuned models (∗ denotes linear finetuning of the DL model)

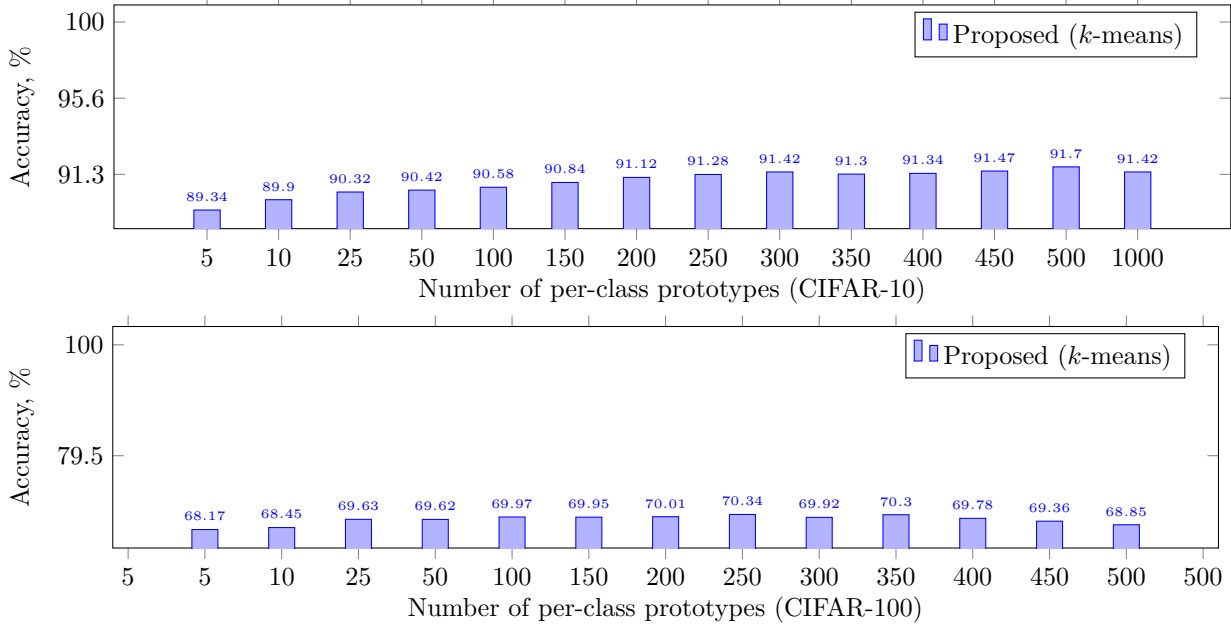

Figure 17: Accuracy sensitivity to the number of per-class prototypes ($k$-means, ResNet101, no finetuning)

| FE | method | accuracy (%) | #prototypes | time, s |
|---|---|---|---|---|
| ResNet50 | random | $41.66 \pm 0.74$ | $1,200(2.4\%)$ | 82 |
| | random | $54.37 \pm 0.43$ | $10,000(20\%)$ | 82 |
| | ELM | $57.94 \pm 0.11$ | $7,524(15.05\%)$ | 129 |
| | xDNN | $58.25 \pm 0.64$ | $884(1.77\%)$ | 98 |
| | $k$-means | $62.67 \pm 0.26$ | $1,200(2.4\%)$ | 124 |
| | $k$-means | $64.07 \pm 0.37$ | $10,000(20\%)$ | 258 |
| ResNet101 | random | $50.25 \pm 0.71$ | $1,200(2.4\%)$ | 128 |
| | random | $61.90 \pm 0.41$ | $10,000(20\%)$ | 128 |
| | ELM | $64.42 \pm 0.12$ | $4,685(9.37\%)$ | 161 |
| | xDNN | $64.60 \pm 0.39$ | $878(1.76\%)$ | 143 |
| | $k$-means | $68.59 \pm 0.40$ | $1,200(2.4\%)$ | 170 |
| | $k$-means | $70.04 \pm 0.12$ | $10,000(20\%)$ | 310 |
| VGG16 | random | $26.16 \pm 0.24$ | $1,200(2.4\%)$ | 94 |
| | random | $37.74 \pm 0.48$ | $10,000(20\%)$ | 94 |
| | ELM | $48.53 \pm 0.05$ | $2,878(5.76\%)$ | 122 |
| | xDNN | $47.78 \pm 0.41$ | $871 \ (1.74\%)$ | 119 |
| | $k$-means | $51.99 \pm 0.24$ | $1,200(2.4\%)$ | 175 |
| | $k$-means | $52.55 \pm 0.27$ | $1,200(2.4\%)$ | 437 |
| ViT | random | $72.39 \pm 0.21$ | $10,000(20\%)$ | 604 |
| | ELM | $69.94 \pm 0.06$ | $8,828(17.66\%)$ | 642 |
| | xDNN | $76.24 \pm 0.24$ | $830(1.66\%)$ | 613 |
| | $k$-means | $79.12 \pm 0.28$ | $10,000(20\%)$ | 673 |
| ViT-L | $k$-means | $\mathbf{82.18 \pm 0.14}$ | $10,000(20\%)$ | $3,905$ |
| | $k$-means (nearest) | $78.75 \pm 0.29$ | $10,000(20\%)$ | $3,909$ |

Table 4: CIFAR-100 classification task comparison for the case of no finetuning of the feature extractor

| FE | method | accuracy (%) | #parameters | #prototypes | time, s | interpretability |
|---|---|---|---|---|---|---|
| ResNet50 | ResNet50 | 79.70 (56.39*) | $\sim 25M$ (205K) | | 36, 360(13, 003*) | ✗ |
| | random | 78.94 ± 0.17 | $\sim 25M + 50K$ | 1, 200(2.4%) | 36, 360 + 28 | ✓ |
| | random | 79.52 ± 0.17 | $\sim 25M + 50K$ | 10, 000(20%) | 36, 360 + 28 | ✓ |
| | xDNN | 79.75 ± 0.12 | $\sim 25M + 50K$ | 859(1.72%) | 36, 360 + 45 | ✓ |
| | k-means | 79.84 ± 0.07 | $\sim 25M + 50K$ | 1, 200(2.4%) | 36,360+82 | ✓ |
| | k-means | 79.77 ± 0.07 | $\sim 25M + 50K$ | 10, 000(20%) | 36,360+219 | ✓ |
| ResNet101 | ResNet50 | 84.38 (63.18*) | $\sim 44M$ (205K) | | 45, 619(18, 955*) | ✗ |
| | random | 82.26 ± 0.15 | $\sim 44M + 50K$ | 1, 200(2.4%) | 45, 619 + 175 | ✓ |
| | random | 80.75 ± 0.19 | $\sim 44M + 50K$ | 10, 000(20%) | 45, 619 + 175 | ✓ |
| | xDNN | 81.13 ± 0.16 | $\sim 44M + 50K$ | 831(1.66%) | 45, 619 + 191 | ✓ |
| | k-means | 83.03 ± 0.06 | $\sim 44M + 50K$ | 1, 200(2.4%) | 45, 619 + 220 | ✓ |
| | k-means | 83.14 ± 0.19 | $\sim 44M + 50K$ | 10, 000(20%) | 45, 619 + 439 | ✓ |
| VGG-16 | VGG-16 | 75.08 (62.74*) | $\sim 138M$ (410K) | | 41, 038(17, 098*) | ✗ |
| | random | 53.83 ± 0.91 | $\sim 138M + 50K$ | 1, 200(2.4%) | 41, 038 + 92 | ✓ |
| | random | 64.17 ± 0.36 | $\sim 138M + 50K$ | 10, 000(20%) | 41, 038 + 92 | ✓ |
| | xDNN | 72.63 ± 0.11 | $\sim 138M + 50K$ | 907(1.81%) | 41, 038 + 120 | ✓ |
| | k-means | 73.83 ± 0.16 | $\sim 138M + 50K$ | 1, 200(2.4%) | 41, 038 + 199 | ✓ |
| | k-means | 73.73 ± 0.23 | $\sim 138M + 50K$ | 10, 000(20%) | 41, 038 + 460 | ✓ |
| ViT | ViT | 90.29(82.79*) | $\sim 86M$ (77K) | | 15, 536(15, 423*) | ✗ |
| | random | 89.90 ± 0.10 | $\sim 86M + 50K$ | 10, 000(20%) | 15, 536 + 621 | ✓ |
| | xDNN | 89.17 ± 0.18 | $\sim 86M + 50K$ | 809(1.61%) | 15, 536 + 630 | ✓ |
| | k-means | 90.48 ± 0.05 | $\sim 86M + 50K$ | 10, 000(20%) | 15, 536 + 695 | ✓ |

Table 5: CIFAR-100 classification task comparison for the case of finetuned models (∗ denotes linear fine-tuning of the DL model)

| FE | method | accuracy (%) | #prototypes | time, s |
|---|---|---|---|---|
| ViT | random | 98.55 ± 0.09 | 500(10%) | 61 |
| | ELM | 95.27 ± 0.03 | 271(5.42%) | 63 |
| | xDNN | 98.63 ± 0.12 | 84(1.68%) | 62 |
| | k-means | 99.32 ± 0.03 | 500(10%) | 65 |
| ViT-L | k-means | 99.71 ± 0.02 | 500(10%) | 377 |
| | k-means(nearest) | 99.56 ± 0.05 | 500(10%) | 377 |

Table 6: STL10 classification task comparison for the case of no finetuning (linear finetuning of the ViT gives 98.97%)

| FE | method | accuracy (%) | #prototypes | time, s |
|---|---|---|---|---|
| ViT | random | 90.82 ± 0.53 | 365(9.92%) | 48 |
| | ELM | 90.85 ± 0.03 | 122(3.32%) | 49 |
| | xDNN | 96.30 ± 0.23 | 239(6.49%) | 49 |
| | k-means | 94.07 ± 0.20 | 365(9.92%) | 50 |
| ViT-L | k-means | 95.78 ± 0.19 | 365(9.92%) | 279 |
| | k-means (nearest) | 94.76 ± 0.30 | 740(9.92%) | 279 |

Table 7: OxfordIIIITPets classification task comparison for the case of no finetuning (linear finetuning of ViT gives 94.41%)

| FE | method | accuracy (%) | #prototypes | time, s |
|---|---|---|---|---|
| ViT | random | $82.67 \pm 0.54$ | $2,154(9.97\%)$ | 266 |
| | ELM | $83.69 \pm 0.01$ | $528(2.44\%)$ | 277 |
| | xDNN | $85.24 \pm 1.05$ | $102(0.47\%)$ | 269 |
| | $k$-means | $91.30 \pm 0.16$ | $2,154(9.97\%)$ | 330 |
| ViT-L | $k$-means | $88.93 \pm 0.22$ | $2,154(9.97\%)$ | 1685 |
| | $k$-means(nearest) | $83.97 \pm 0.16$ | $2,154(9.97\%)$ | 1685 |

Table 8: EuroSAT classification task comparison for the case of no finetuning (linear finetuning gives 95.17%)

| FE | method | accuracy (%) | #prototypes | time, s |
|---|---|---|---|---|
| ViT | random | $89.42 \pm 0.32$ | $649(9.35\%)$ | 96 |
| | ELM | $91.12 \pm 0.07$ | $516(7.43\%)$ | 97 |
| | xDNN | $94.61 \pm 0.94$ | $579(8.34\%)$ | 97 |
| | $k$-means | $94.46 \pm 0.44$ | $649(9.35\%)$ | 99 |
| ViT-L | $k$-means | $96.08 \pm 0.34$ | $649(9.35\%)$ | 515 |
| | $k$-means (nearest) | $93.74 \pm 0.42$ | $649(9.35\%)$ | 517 |

Table 9: CalTech101 classification task comparison (linear finetuning gives 96.26%)

First, we show the symbolic decision rules in Figure 18. These symbolic rules are created using ViT-L backbone, with the prototypes selected using the nearest real image to $k$-means cluster centroids, in a no-finetuning scenario for OxfordIIITPets dataset.

Second, in Figure 19 we show how the overall pipeline of the proposed method can be summarised in interpretable-through-prototypes fashion. We show the normalised distance obtained through dividing by the sum of distances to all prototypes. This is to improve the perception and give relative, bound between 0 and 1, numbers for the prototype images.

Figure 18: An example of symbolic decision rules (OxfordIIITPets), $Q$ denotes the query image

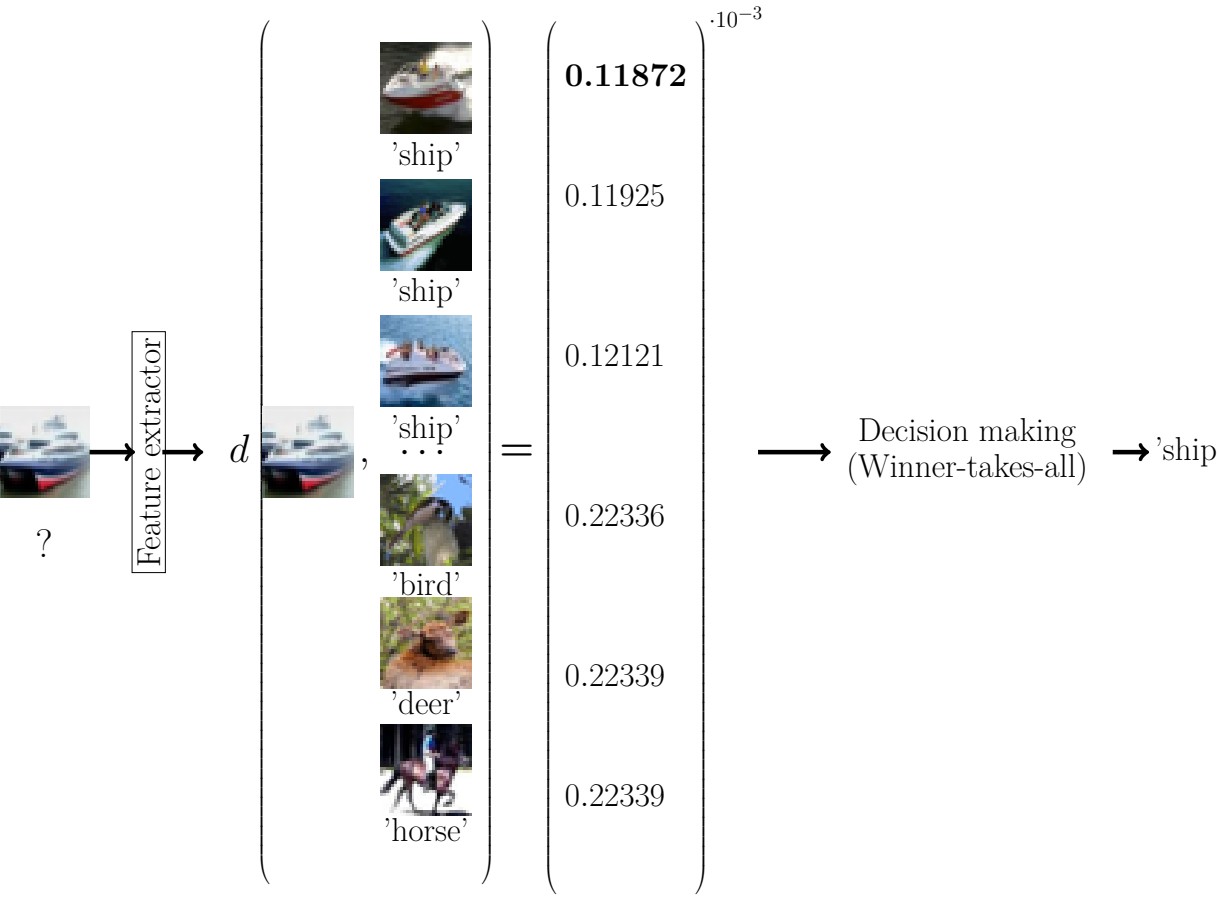

Figure 19: Interpreting the model predictions ($k$-means (nearest), 500 clusters per class, CIFAR-10, ViT)

