# OpenReview forum: "IDEAL: Interpretable-by-Design ALgorithms for learning from foundation feature spaces"
_TMLR — Rejected by TMLR_

### Review · Reviewer_77n3 · 2023-12-16

**Summary Of Contributions:**

In this paper, the authors notice that current existing deep learning methods that rely on parametric tuning lack explanatity. Based on this observation, they propose a framework which is called IDEAL in order to solve the aforementioned problem. Briefly speaking, the IDEAL algorithm recasts the conventional supervised classification problem into a function of similarity to a set of prototypes derived from training data while taking advantage of existing latent space of foundation models. It is claimed that the proposed IDEAL method is interpretable via prototypes while also mitigating the issue of confounding bias and able to circumvent the issue of catastrophic forgetting. Further, the proposed IDEAL method also narrows the gap between finetuned and non-finetuned models allowing for transfer learning in a fraction of time without finetuning of the features space on a target dataset.

**Audience:**

No

**Claims And Evidence:**

No

**Requested Changes:**

- The topic of the paper should be narrowed to transfer learning/fine-tuning pretrained models instead of using deep learning, which is a broad domain.
- The formulations, such as Eq. (1), requires more polish work.
- More explanations and theoretical analyses are required to reveal that IDEAL method is interpretable.
- More works are required to improved the contribution of this paper.

**Strengths And Weaknesses:**

__Cons:__
- From my perspective, the paper mainly focuses on fine-tune of a given pre-trained model, since the main concerns mentioned in this paper are closely related to transfer learning (e.g. the 3 limitations mentioned in Section 1). Thus, it is better to constrain the topic in transfer learning instead of the entire deep learning.
- The formulations of this paper are not clear and comprehensive.
- The background part of this paper is not clear. The motivation is lacked here. It is hard to understand why Eq. (1) has the 3 limitations and why this problem is worth studying.
- The paper claims that interpretable explanations are provided. However, I do not see any of them before experiment section.
- The novelty of this paper is poor. In fact, prototypical learning paradigm has been widely used in many previous works, such as prototypical networks proposed in few-shot learning problem. Compared with previous works like prototypical nets that directly leverage label information to generate prototypes, this paper selects samples from the training data. I do not think such strategy is better.

__Questions:__
1. It seems that Eq. (1) is not very clear to express the learning process of a model. To be specific, the notation of $f(x|\theta)$ is confusing. It somewhat resembles the expression of conditional probability. However, we usually express the encoding process of a model in the way of $f_{\theta}(x)$.
2. Since $d$ is a similarity function, what is $h$? (see Eq. (3))
3. We do not see some content regarding interpretability in this paper. Thus, how do you demonstrate the claimations in section backgound?

__Minor concerns:__
1. In Fig. 1(a), there is a $f_2(\cdot|\theta_1)$. I think it should be $f_2(\cdot|\theta_2)$.

---

> ### Author Response · Authors · 2024-01-15
> **Rebuttal**
>
> Thank you for the review. Emphasis on transfer learning is indeed a good idea. We are going to reflect it in the revised version of the paper due to appear this week. We also realise that a title may look overstated, and a better name could be selected for the paper. We propose, therefore, to rename the paper “IDEAL: Interpretable-by-Design ALgorithms for learning from foundation feature spaces”.
>
> *“The novelty of this paper is poor. In fact, prototypical learning paradigm has been widely used in many previous works, such as prototypical networks proposed in few-shot learning problem. Compared with previous works like prototypical nets that directly leverage label information to generate prototypes, this paper selects samples from the training data. I do not think such strategy is better."*
>
> We do not think this comment duly reflects the contributions and contents of the paper.  We also do not think that the journal guidelines scope it out this way. The criteria explicitly emphasise interest over novelty: https://www.jmlr.org/tmlr/acceptance-criteria.html *“Nor should it form the basis for rejecting work on a method considered not “novel enough”, as novelty of the studied method is not a necessary criteria for acceptance. We explicitly avoid these terms (“significant”, “impactful”, “novel”), and focus instead on the notion of “interest”. If the authors make it clear that there is something to be learned by some researchers in their area from their work, then the criterion of interest is considered satisfied”*  Below, we clarify why we think this work is of interest to the community (and show what research gap we are addressing).
>
> We do *not* claim to propose the prototypical learning paradigm, and indeed, a number of existing methods are presented in Section 2. Using samples from training data is a common strategy which is used in both ProtoPNet (Chen et al, 2019) and xDNN (Angelov&Soares, 2020). Instead, we see the key contribution to be the experimental analysis. To enable such analysis, we propose a framework, inspired by prototypical models such as ProtoPNet and xDNN, and place the novelty on empirical findings demonstrating that:
> - vision transformer backbones decrease the gap between finetuned and non-finetuned models (Section 4.2 and 4.3)
> - one can competitively solve incremental learning problems (Section 4.4)
> - the model allows to interpret the decision making (Section 4.5)
> - finally and importantly, that finetuning is prone to confounding bias, and models without finetuning can outperform the purpose-finetuned ones in the proposed setting (Section 4.6).
>
> We argue that neither of these experimental findings, which can have value for the wider transfer learning community, have been previously described. Section 4.6 shows that non-finetuned models, in certain  scenarios, can outperform purpose-finetuned ones. We will make changes to better reflect this motivation in the first two sections of the paper.
>
> *“The formulations of this paper are not clear and comprehensive.”* It would be much appreciated if the reviewer could be more specific about this, however, we are planning to improve upon the motivation and clarity according to the comments in the upcoming revision.
>
> *“The paper claims that interpretable explanations are provided. However, I do not see any of them before experiment section.”*
> We consider the setting of interpretability in the latent space of the deep learning models, and we put forward the empirical argument, and therefore we think it belongs to the empirical section. To be more precise, the claims of interpretations are addressed in questions 4 and 5 of the experimental section:
>
> **Q4: How does the IDEAL framework provide insight and interpretation?**
>
> **Q5: Can models without finetuning bring advantage over the finetuned ones in terms of accuracy and help identify misclassifications due to confounding (i.e., spurious correlations in the input)?**
>
> *"The formulations, such as Eq. (1), requires more polish work."*
> Thank you, we are going to fix it the update for the revised version.
>
> *"More explanations and theoretical analyses are required to reveal that IDEAL method is interpretable."*
> We put the explainations into the experimental section because we focus on the empirical aspects of explainability, which is the standard practice in many peer-reviewed papers on the topic as described in Section 2, paragraph “Explainability and Interpretability”. In the revised version of the background section, we will link the state-of-the-art understanding of theory of causal and disentangled learning with the motivation of the paper. To address your comments and those by the reviewer 2uCm, we have prepared additional analysis, uncovering the reasons behind improvements in the scenarios without confounding (see the revised version of the paper). We are also working at the moment to address the concerns on clarity and will release it later during the week.

---

> ### Author Response · Authors · 2024-01-23
>
> Further to the previous discussion, we:
> - improved the notation of the paper, including changing the notation from $f(\cdot| \theta)$ to $f(\cdot ; \theta)$, following the example from existing well-known works such as [1], and added clarifications around Eq (1)
> - we fixed the reported typos
> - we narrowed the topic down to interpretable transfer learning to better reflect the context of the paper; we changed the title accordingly
> - we addressed the reported issues of clarity and emphasised the contribution, as well as contrasted our work with the related prototype-based methods
>
> Thank you again for your feedback, and looking forward to hearing back from you.
>
> [1] Papamakarios, G., Nalisnick, E., Rezende, D. J., Mohamed, S., & Lakshminarayanan, B. (2021). Normalizing flows for probabilistic modeling and inference. The Journal of Machine Learning Research, 22(1), 2617-2680.

---

### Review · Reviewer_2uCm · 2023-12-20

**Summary Of Contributions:**

The study introduces IDEAL (Interpretable-by-design DEep learning ALgorithms), a framework enhancing deep learning (DL) explainability and addressing limitations of existing models like ProtoPNet and xDNN. IDEAL transforms supervised classification into a similarity function with prototypes derived from training data, utilizing latent spaces of large neural networks. Key findings include: IDEAL's interpretability through prototypes, mitigation of confounding bias, avoidance of catastrophic forgetting in class-incremental learning, and efficient transfer learning without finetuning feature space. The study demonstrates these advantages over traditional models across multiple datasets​​.

**Audience:**

Yes

**Claims And Evidence:**

No

**Requested Changes:**

1. The paper could benefit from clearer organization and more precise language to enhance readability and understanding, particularly in explaining complex concepts and comparisons with existing models.

2. Given the strengths and weaknesses identified, the paper shows promise but requires significant revisions to address the weaknesses, particularly in clarifying its unique contributions and improving the overall clarity and coherence of the manuscript.

**Strengths And Weaknesses:**

Strengths
1. Comprehensive Experiments: The study conducts extensive experiments across a variety of datasets (CIFAR-10, CIFAR-100, CalTech101, STL-10, Oxford-IIIT Pet, EuroSAT), demonstrating the effectiveness of the IDEAL framework in different contexts.

2. Improved Interpretability: The use of prototypes in the IDEAL framework enhances the interpretability of deep learning models, a critical aspect often overlooked in traditional deep learning approaches.

Weaknesses
1. Unclear Mitigation of Confounding Bias: The paper does not sufficiently clarify how the proposed IDEAL method mitigates the issue of confounding bias, leaving a gap in understanding its effectiveness in this regard.

2. Lack of Distinction from Existing Models: There is a lack of clarity on how IDEAL significantly differs from existing prototype-based models like ProtoPNet and xDNN, especially regarding its claim of going beyond end-to-end learning and utilizing large classifiers' feature space.

3. Overstated Title: The title of the paper suggests a broader scope than what is presented. The focus on lifelong/continuing learning in the title does not accurately reflect the core content and contributions of the proposed framework.

4. Interpretability of Models vs. Predictions: The paper emphasizes the interpretability of predictions but does not adequately address the interpretability of the models themselves, which is a more crucial aspect in the context of explainable AI.

---

> ### Author Response · Authors · 2024-01-15
>
> Dear Reviewer 2uCm,
>
> Many thanks for the constructive comments which will help improve our paper in the next revision.
>
> Please find below our answer on your questions:
>
> *“Unclear Mitigation of Confounding Bias: The paper does not sufficiently clarify how the proposed IDEAL method mitigates the issue of confounding bias, leaving a gap in understanding its effectiveness in this regard.”*
> We think this is an excellent question. To answer it, we have prepared new analysis which investigates the impact of the confounding bias on the representations in finetuned and non-finetuned feature spaces and underpins the reasons behind the model’s effectiveness comparing to the finetuned scenario. Please find it attached in the updated paper (last paragraph of Page 14, and Figure 17 of the updated paper). We are working on improving the clarity of the paper in accordance with the comments in the meantime.
>
> *“Overstated Title: The title of the paper suggests a broader scope than what is presented. The focus on lifelong/continuing learning in the title does not accurately reflect the core content and contributions of the proposed framework.”*
> We agree with this, and we are planning to change the title to “IDEAL: Interpretable-by-Design ALgorithms for learning from foundation feature spaces”, as well as better reflect the place of the method within transfer learning literature. Please let us know if you are happy with that.
>
> *“Interpretability of Models vs. Predictions: The paper emphasizes the interpretability of predictions but does not adequately address the interpretability of the models themselves, which is a more crucial aspect in the context of explainable AI."*
> Admittedly, the limitation of this setting is that we focus on interpretability of the predictions and not the models. We think that this limitation can be potentially mitigated by using this framework in conjunction with interpretable-by-design architectures such as, for example, B-Cos (Bohle et al, 2022). We will expand upon these limitations and the ways to mitigate them, in the limitations section of the paper.
>
> We also realise from this comment that we need to put it in the wider context of interpretability research. For this purpose, in the revised version of the paper, we will link it in the related works section with the wider literature of interpretability of models, drawing upon the links with research on causality, disentangled learning and robustness.
>
> *“The paper could benefit from clearer organization and more precise language to enhance readability and understanding, particularly in explaining complex concepts and comparisons with existing models.”*
>
> *“Lack of Distinction from Existing Models: There is a lack of clarity on how IDEAL significantly differs from existing prototype-based models like ProtoPNet and xDNN, especially regarding its claim of going beyond end-to-end learning and utilizing large classifiers' feature space.”*
>
> Many thanks! We are now working on improving the clarity of the paper, especially the introduction and on contrasting with existing models. We will emphasise that while xDNN and ProtoPNet are based on end-to-end finetuning of the backbone, we focus our analysis on the no-finetuning prototypical scenario.
>
> *“Given the strengths and weaknesses identified, the paper shows promise but requires significant revisions to address the weaknesses, particularly in clarifying its unique contributions and improving the overall clarity and coherence of the manuscript.”*
> We are now working on revising the manuscript in improving its coherence and clarity as proposed. We see the unique contributions of the paper in its empirical evaluation, quantifying the benefits of no-finetuning scenarios for a number of settings such as transfer and lifelong learning, and providing human-interpretable analysis of decision making in such models. We would like to particularly single out the contribution of the confounding bias experiment (Question 5) which shows that in the controlled scenarios with confounding bias, non-finetuned models can outperformed the purpose-finetuned ones.

---

> > ### Author Response · Authors · 2024-01-23
> >
> > We address the requested changes as outlined in the general response.
> >
> > We have been focused on improving clarity, including changing the title, abstract and the background section, as well as providing better context for the distinction from the related methods and improved method description.
> >
> > We have also performed, as we have updated you last week, the new experiment to show the impact of confounding  (Page 15 and Figure 16 of the current version) as detailed below.
> >
> > Please let us know what you think.

---

### Review · Reviewer_Pvd6 · 2024-01-12

**Summary Of Contributions:**

The paper proposes a new interpretable neural network method named IDEAL. Like xDNN and ProtoPNet, IDEAL uses prototypes to make network interpretable and doesn't require fine-tuning on downstream tasks since it can use k-NN classifier. Experiments show the effect of IDEAL.

**Audience:**

No

**Broader Impact Concerns:**

Since it is hard to understand the contribution and novel points of the proposed method, the paper is not ready to consider broader impact concerns.

**Claims And Evidence:**

No

**Requested Changes:**

- It is hard to understand what IDEAL is. I request to define the proposed method IDEAL precisely.
- After defining IDEAL, compare it with previous approaches such as xDNN, ProtoPNet, and DNC with formulas and categorizations.
- Compare the performance of IDEAL with previous methods in the experiments section.
- Add explanation and experiments for why k-NN classifier is novel in IDEAL.

**Strengths And Weaknesses:**

### Strengths
- The paper reports various experiment results

### Weakness
- It is hard to understand the differences and advantages of IDEAL from previous approaches
  - There are a lot of papers on prototype-based interpretable models. But, no formula or summary explains novel points of IDEAL compared to others.
  - Experiments look self-comparison with the prototype selection method from xDNN. Does IDEAL outperforms other methods in performance or interpretability measure?
- **without finetuning** setting looks similar to k-NN classifier, which is not novel in DNN classification. Is there any special reason that other methods can't be used for k-NN classifier?
- Method section needs to be improved with details
  - I recommend authors to move eq (7), (8) to the method section. It is hard to understand d and h without this formula at method section.
  - I need more information related to Algorithm 1,2. What are the details of functions `FindPrototype`, `SelectParameters`, `UpdateParameters`, and `UpdatePrototype`?

---

> ### Author Response · Authors · 2024-01-15
>
> We would like to make a number of clarifications, as we are not sure the contributions actually summarise the paper.
>
> 1. On the question of novelty: we  don’t think the journal guidelines scope it out this way, The criteria explicitly emphasise interest over novelty: https://www.jmlr.org/tmlr/acceptance-criteria.html *“Nor should it form the basis for rejecting work on a method considered not “novel enough”, as novelty of the studied method is not a necessary criteria for acceptance. We explicitly avoid these terms (“significant”, “impactful”, “novel”), and focus instead on the notion of “interest”. If the authors make it clear that there is something to be learned by some researchers in their area from their work, then the criterion of interest is considered satisfied”*
>
>  Below, we clarify why we think this work is of interest to the community (and show what research gap we are addressing).  One of the motivations behind foundation models is the decrease of the gap between finetuned and non-finetuned models to the extent that it one can avoid finetuning altogether. While the notion of it is well-known, there is not enough analysis available in quantifying such gap. In this work, we propose a setting for prototype-based interpretable learning over such foundation models, inspired by xDNN, referred to as IDEAL, and quantify the gap between finetuned and non-finetuned models in this setting. The main contribution, therefore, is empirical quantification of the trade-offs between the finetuned and non-finetuned models. We hope that this extensive empirical analysis could help understand the trade-offs between finetuning and non-finetuning of backbones and can be of interest to the community. Importantly, in contrast to the common wisdom that the non-finetuned models are bound to trail behind the finetuned ones, we show that in the case of confounders (Section 4.6), a model without finetuning can surpass the performance of finetuned models. We believe that this empirical analysis can be of interest to the community. More precisely, the contribution is summarised in a number of empirical findings:
> * vision transformer backbones decrease the gap between finetuned and non-finetuned models (Section 4.2 and 4.3)
> * without finetuning, one can competitively solve incremental learning problems (Section 4.4)
> * the framework allows to interpret the decision making process (Section 4.5)
> * finally and importantly, that finetuning is prone to confounding bias, and, against the conventional wisdom that non-finetuned models cannot beat the purpose-fine tuned ones, models without finetuning can **outperform** the purpose-finetuned ones in the setting with confounders (Section 4.6).
>
> 2. *“The paper proposes a new interpretable neural network method named IDEAL. Like xDNN and ProtoPNet, IDEAL uses prototypes to make network interpretable and doesn't require fine-tuning on downstream tasks since it can use k-NN classifier. Experiments show the effect of IDEAL.”* We don’t think this summary is fair (see Figure 3). First of all, it is not a k-NN classifier, it is prototype selection (through clustering) with winner-takes-all (1-NN, or, in an ablation study, k-NN) decision making. Second, the contribution is not proposing a yet another novel method, but to quantify the performance in the non-finetuned foundational model setting, inspired by xDNN, to show the trade-offs between finetuning and not finetuning the foundation models, in terms of performance in online and offline learning scenarios, as well as interpretability of decision making.  To make such setting possible, we needed to define the framework  which allows us to carry out such comparison.
>
> 2. *“There are a lot of papers on prototype-based interpretable models. But, no formula or summary explains novel points of IDEAL compared to others.”*
>
> *“without finetuning setting looks similar to k-NN classifier, which is not novel in DNN classification. Is there any special reason that other methods can't be used for k-NN classifier?”*
> First, we think it would be prototype selection+k-NN and not k-NN. There is indeed a long history of works as we describe in Section 2, including, notably, Snell et al (2017), Chen et al (2019), Angelov & Soares (2020). We will stress the differences between these models in the revised version.
>
> 4. *“It is hard to understand what IDEAL is. I request to define the proposed method IDEAL precisely. After defining IDEAL, compare it with previous approaches such as xDNN, ProtoPNet, and DNC with formulas and categorizations.”*
>  We will better contrast IDEAL with well-known methods as suggested and revise the clarity of the framework definition.
>
> 5. *“Compare the performance of IDEAL with previous methods” * As we stated before, our primary contribution is understanding the trade-offs between finetuning and no finetuning in the prototypical setting, which necessitates definition of the framework, and not in proposing a better prototypical model.

---

> > ### Author Response · Authors · 2024-01-23
> >
> > We address the requested changes as outlined in the general response.
> >
> > "It is hard to understand what IDEAL is. I request to define the proposed method IDEAL precisely."
> > To follow upon the previous comments, we have revised the method description, as well as the background, to improve the definition of the methodology.
> >
> > "Compare the performance of IDEAL with previous methods in the experiments section."
> > We outline, in Section 3.3, how the proposed methodology relates to the state-of-the-art contributions: xDNN, ProtoPNet, and DNC. Section 4 explicitly compares with the prototype selection in xDNN.
> >
> > Please let us know what you think about the revised version.

---

### Review · Reviewer_wmNR · 2024-01-18

**Summary Of Contributions:**

The paper proposes a framework called IDEAL (Interpretable-by-design DEep learning ALgorithms) which recasts the standard supervised classification problem into a function of similarity to a set of prototypes derived from the training data while taking advantage of existing latent spaces of large neural networks forming so-called Foundation Models (FM).

**Audience:**

Yes

**Broader Impact Concerns:**

The proposed research has the potential to advance the field of interpretable deep learning and provide more transparent and trustworthy models for various applications, the authors could consider this as well. Also, they could add possible negative impacts of their framework, such as the misuse or abuse of the prototype-based explanations, the privacy and security risks of exposing the prototypes to the users, or the ethical and social implications of using large foundation models that may encode biases or prejudices. I do not see any major concern regarding the broader impact of this paper, but I encourage the authors to address these issues in their future work.

**Claims And Evidence:**

Yes

**Requested Changes:**

- The authors need to add an introduction and a coherent story at the beginning of the paper.
- Algorithms 1 needs to include more information on how the method works.
- Authors need to compare their method with previous work and provide the novelty of this work explicitly.

**Strengths And Weaknesses:**

**Strengths**:
- The paper proposes a novel and generic framework called IDEAL that can be applied to various deep learning architectures and datasets with/without fine-tuning and demonstrates its advantages over existing methods in terms of performance, interpretability, and transfer and lifelong learning.
- The paper provides a comprehensive evaluation of the proposed framework on multiple datasets and tasks and shows that it can achieve good results without finetuning the feature space, handle class-incremental learning scenarios, and generate prototype-based explanations and symbolic rules.

**Weaknesses**:
- The paper does not provide a clear introduction to an unfamiliar audience. Also, the background second is not coherent and does not define what is the proposed method and jumps to the "efficiency of the proposed method".
- The paper provides several figures that provide a clear explanation of the proposed method but it would be great to reduce the redundancy and provide one figure whit a visual abstract of the work.

---

> ### Author Response · Authors · 2024-01-23
>
> Many thanks for your response and feedback. We have revised the paper accordingly as we detail in the common comment to all reviewers.
>
> This includes:
> - updating the introductory part to improve the coherence of the story, as well as improving upon the motivation
> - updating the algorithms and their description, as well as contrasting the model with other prototypical methods: xDNN, ProtoPNet, DNC
> - removing the figure from Section 3.4 to make the description more concise

---

### Author Response · Authors · 2024-01-22
**Summary of the new revision**

The new revision addresses the number of points regarding clarity, soundness and contribution. The changes are marked in red in the draft and summarised below:
- to address a number of questions regarding the motivation of the paper, we updated both the abstract and the introduction. Now it emphasises that the paper addresses the challenge of interpretable transfer learning in nascent foundation feature spaces, and how the paper addresses this challenge through definition of the framework and the experiments.
- we emphasise how the proposed framework is related to other prototypical learning models, and demonstrates, in Section 3.3, how IDEAL generalises these models
- throughout the paper, we improve the notation to address comments from Reviewer 77n3
- we revised the formulation of the Algorithms 1 and 2 to improve upon clarity
- we removed the graphic summary from Section 3.4 to enable more concise description as per suggestion by Reviewer wmNR
- We moved the discussion about choice of dissimilarity d and decision-making function h into the Section 3.2, and added a new section on the difference from other prototypical frameworks in Section 3.3 as per request by Reviewer Pvd6.
- In Section 4.6, we added new experiment to give an intuition about how the models without finetuning perform better in a confounding scenario than purpose-fine tuned ones (see discussion around Figure 16)
- We changed the title, as we agree with Reviewers 77n3 and 2uCm that the previous one was overstated
- We updated broad impact statement, as per request by Reviewer wmNR

---

### Decision · Action_Editor_9wDB · 2024-02-13

**Recommendation:** Reject

**Comment:**

The paper is motivated to address an issue of no interpretation for many learning algorithms, in particular for deep transfer learning. The paper then moves to make use of pre-trained foundation models and proposes a framework called IDEAL (Interpretable-by-design DEep learning ALgorithms). IDEAL leverages the powerful foundation models for feature extraction, and is thus expected to be better than previous prototype learning methods. The paper conducts a series of experiments analyzing how the choices of latent space, prototype selection, and finetuning of the latent space affect accuracy and generalisation of the models on transfer learning scenarios for different backbones. The paper also mentions a few other benefits possibly brought by the proposed IDEAL.

Three reviewers give thorough reviews on the paper. They all have common concerns on selling points of the paper - motivation and organization of the paper are less clear such that it is difficult to evaluate what are the key messages from the paper. It is also difficult to identify advantages of the proposed IDEAL compared with previous prototype learning methods. They also raise issues about technical writing, e.g., the confusing notations of f(x|\theta), eqn.(3), etc. The authors are expected to narrow down the paper scope, propose clearer technical contributions, and compare with the closely related methods.

**Audience:**

Yes, people focusing on transfer learning and feature learning would be interested in the studied topics.

**Claims And Evidence:**

No. The claims are less clear, and supporting evidence is not convincing.